# Contrastive Localized Language-Image Pre-Training

## Abstract

Contrastive Language-Image Pre-training (CLIP) has been a celebrated method for training vision encoders to generate image/text representations facilitating various applications. Recently, CLIP has been widely adopted as the vision backbone of multimodal large language models (MLLMs) to connect image inputs for language interactions. The success of CLIP as a vision-language foundation model relies on aligning web-crawled noisy text annotations at *image levels*. Nevertheless, such criteria may become insufficient for downstream tasks in need of fine-grained vision representations, especially when *region-level* understanding is demanding for MLLMs. In this paper, we improve the localization capability of CLIP with several advances. We propose a pre-training method called **C**ontrastive **Loc**alized Language-Image Pre-training (**CLOC**) by complementing CLIP with region-text contrastive loss and modules. We formulate a new concept, *promptable embeddings*, of which the encoder produces image embeddings easy to transform into region representations given spatial hints. To support large-scale pre-training, we design a visually-enriched and spatially-localized captioning framework to effectively generate region-text pseudo-labels at scale. By scaling up to billions of annotated images, CLOC enables high-quality regional embeddings for image region recognition and retrieval tasks, and can be a drop-in replacement of CLIP to enhance MLLMs, especially on referring and grounding tasks.

## 1 Introduction

Large-scale vision-language (VL) pre-training has been an important foundation for the recent tremendous growth of multimodal applications. Contrastive Language-Image Pre-training (CLIP) (Radford et al., 2021; Jia et al., 2021) has become a great success of VL representation learning that connects images and text by contrastive training on web-crawled image-text pairs. It has been proven strong transferability and generalizability on extensive downstream tasks such as zero-shot image classification and image-text retrieval. Even beyond, CLIP has become arguably the default choice of vision backbone for multimodal large language models (MLLMs) (Liu et al., 2023; Achiam et al., 2023; McKinzie et al., 2024) due to its superior prior knowledge in aligning vision and language (Tong et al., 2024), facilitating vision inputs to be injected into language models.

As VL research gets increasing attention and expedites progress, various more advanced multimodal tasks are demanding stronger vision capabilities. For instance, recent MLLMs (Rasheed et al., 2024; Ren et al., 2024; Lai et al., 2023; Chen et al., 2023; Peng et al., 2023) have been focusing on more fine-grained understanding tasks that require comprehension of the semantic at *region levels* such as visual question answering (VQA) with referring and grounding instructions. These MLLMs are fine-tuned on referring and grounding data with CLIP as the vision backbone, as seen in works like Kosmos-2 (Peng et al., 2023) and Ferret (You et al., 2023; Zhang et al., 2024). Due to the need for such region-level understanding, CLIP, which aligns entire images with text captions, seems insufficient, as its regular image-text contrastive loss primarily emphasizes global semantics.

To remedy such core localization capability for CLIP, we ask a challenging and fundamental question: *without sacrificing CLIP's original strong image-level knowledge, can we pre-train a stronger image encoder with enhanced localization capability that can be inherently integrated into MLLMs?*

To this end, we explore a data-driven approach that complements the original CLIP image-text pre-training objective with explicit region-text supervision. Though conceptually simple, several

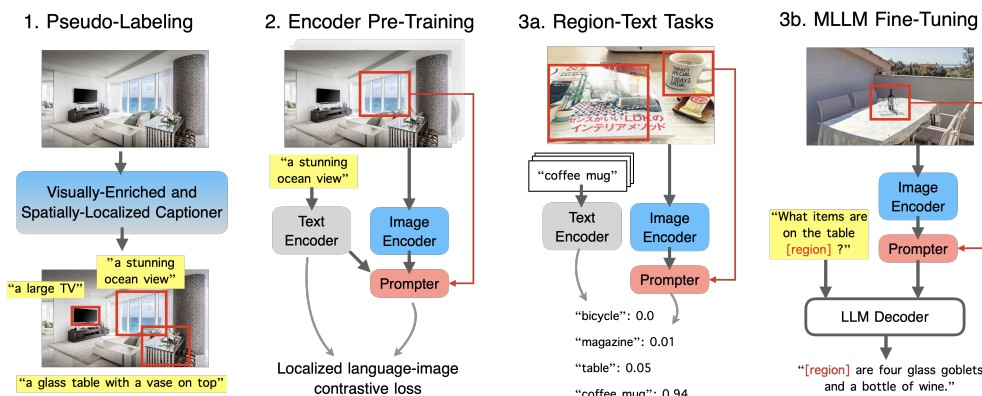

Figure 1: **Overview of our CLOC pre-training framework. (1)** A visually-enriched and spatially-localized captioning pipeline generates pseudo-labeled bounding boxes with detailed descriptions for key image regions. **(2)** A lightweight `Prompter` attached on top of the CLIP image encoder can be prompted to transform the image embedding into the region-focused feature. All parameters are trained end-to-end from scratch with our contrastive localized language-image loss on the annotated region-text datasets. After pre-training, **(3a)** region features can be generated via the `Prompter` for region-text tasks like object classification in a training-free fashion. **(3b)** The image encoder, along with the optional `Prompter`, can also strengthen MLLMs fine-tuning by enhancing their fine-grained image understanding capabilities.

challenges exist. *First*, it lacks public datasets with region-text annotations at scales large enough for CLIP training, which typically requires hundreds of millions even billions of images. Existing region-text corpus like Visual Genome (Krishna et al., 2017) contains about 108K images, and the largest noisy-labeled grounded dataset GRIT (Peng et al., 2023) features only around 20M images. Indeed, such deficiency of labeled datasets has probably limited the literature to mainly consider semi-supervised or weakly-supervised approaches as somewhat a compromise (Naeem et al., 2023; Yao et al., 2022; 2023a).

*Second*, a plausible solution is to scale up training data in pursuit of image regions pseudo-labeled with text annotations via some open-vocabulary detectors (Minderer et al., 2024; Zhang et al., 2022). Though it seems feasible, we found it non-trivial to design such a pipeline as the annotations are noisy and will greatly affect the final model performance. *Third*, even if the region-text datasets are given, it is under-explored how to effectively train on them in terms of co-designs of training objectives, model architecture, and more design details.

To this end, we propose a new pre-training framework illustrated in Figure 1, named **C**ontrastive **Loc**alized Language-Image Pre-Training (**CLOC**), to improve CLIP with better localization capability, especially for MLLMs, by overcoming the above difficulties. Our main contributions are:

- We propose a new learning goal, **Promptable Embeddings**, that *a strong vision encoder should produce image embeddings that can be easily transformed into region representations, given some spatial hints (e.g., box referring or text prompts)*. This formulation not only facilitates the encoder as a prior of fine-grained VL alignment, but also enables new possibilities for the interactions between the image encoder and the language decoder.
- To optimize towards the goal, we design simple and minimal modifications on top of CLIP. We augment the original CLIP loss with a region-text contrastive loss, where the region embeddings are extracted from the image embedding by a lightweight extractor module conditioned on the spatial hints (*i.e.*, *prompts*).
- We design a large-scale pseudo-labeling data engine to support CLOC training. We properly combine visual-enriched image captioners and open-vocabulary detectors for an effective recipe that improves previous practice of region annotations (Minderer et al., 2024; Peng et al., 2023). This approach yields a two-billion image-text dataset with fine-grained region-text annotations, which serves as the foundation for training our CLOC model.
- Through extensive experiments across 31 evaluation tasks, including standard image-text tasks, newly constructed region-text tasks, and downstream evaluations with MLLMs, we demonstrate that CLOC significantly and consistently outperforms the CLIP counterpart.
- We are working on releasing our pre-trained checkpoints and the constructed region-text annotations along with the final version to accelerate future research within the community.

## 2 RELATED WORK

**Improving localization of CLIP.** Since CLIP was introduced, many follow-up works have been proposed to improve it from various aspects, for different target tasks, and with different approaches. From the aspect relevant to our work, improving the localization capability, most works specifically focus on the downstream dense vision tasks such as open-vocabulary detection (Minderer et al., 2024; Yao et al., 2022; Wu et al., 2023). Another less and arguably more challenging thread is to maintain the generalizability of CLIP on image-level tasks while improving localization. Recent works like SILC (Naeem et al., 2023) and SPARC (Bica et al., 2024) combine localization-enhancing unsupervised objectives with the CLIP loss, but do not attempt with supervision on large-scale explicit pseudo-labeled data like ours. Alpha-CLIP (Sun et al., 2024) shows that the SAM segmentation model (Kirillov et al., 2023) can provide useful conditions for CLIP.

**Vision encoder pre-training for MLLMs.** Building upon the success of large language models (LLMs), a popular approach to MLLMs like LLaVA (Liu et al., 2023), typically connects a vision encoder (*e.g.*, ViT (Dosovitskiy et al., 2021)) to digest visual inputs and maps them to the LLM decoder input space as token embeddings. Among various types of vision encoders (Oquab et al., 2023; He et al., 2022), CLIP (Radford et al., 2021; Jia et al., 2021) becomes the most popular choice, due to its superior performance on MLLM benchmarks reported by recent studies (Tong et al., 2024).

**Synthetic annotations for pre-training.** Large-scale training data are the fuel of pre-training, especially for CLIP. The literature has been exploring scalable ways to generate high-quality synthetic annotations. For instance, several works demonstrate that visually-enriched image captions improve CLIP (Lai et al., 2024). MOFI (Wu et al., 2024) constructs a large alt-text set and augments CLIP with a multi-classification task. However, these works only consider image-level annotations but not explicit region-level labels. In the context of dense vision tasks like open-vocabulary detection and segmentation, pseudo-labeling in a self-training paradigm has proven an effective approach (Kirillov et al., 2023; Minderer et al., 2024). We are inspired by these efforts and build on them to enhance CLIP's localization capabilities.

## 3 CLOC: CONTRASTIVE LOCALIZED LANGUAGE-IMAGE PRE-TRAINING

### 3.1 PRELIMINARY: FROM IMAGE-TEXT TO REGION-TEXT ALIGNMENT

Contrastive Language-Image Pre-training (CLIP) (Radford et al., 2021) trains a pair of image and text encoders (denoted as $f_I$ and $f_T$, respectively) by contrastively aligning the image and text embeddings. Let a mini-batch of $N$ image-text pairs $\{(\boldsymbol{x}_i, \boldsymbol{y}_i)\}_{i=1}^N$ be sampled from the large-scale training set during each training iteration. The contrastive loss is defined as follows:

$$\mathcal{L}_{\text{CLIP}} := (\mathcal{L}_{I \to T} + \mathcal{L}_{T \to I})/2. \; \mathcal{L}_{I \to T} := -\frac{1}{N} \sum_{i=1}^N \log \frac{\exp\left(\texttt{sim}(f_I(\boldsymbol{x}_i), f_T(\boldsymbol{y}_i))/\tau\right)}{\sum_{j=1}^N \exp\left(\texttt{sim}(f_I(\boldsymbol{x}_i), f_T(\boldsymbol{y}_j))/\tau\right)}, \quad (1)$$

where $\texttt{sim}(\cdot, \cdot)$ is the similarity measurement function and $\tau$ is a (learnable) logit temperature. The CLIP loss $\mathcal{L}_{\text{CLIP}}$ averages the symmetrical contrastive loss in which cross-entropy normalized along image-to-text and text-to-image axes, respectively.

Conceptually, the CLIP loss aligns images with their associated text, but it overlooks regional information and spatial semantics. We propose augmenting this with *region-text* alignment on top of $\mathcal{L}_{\text{CLIP}}$. Specifically, assume an image-text pair $(\boldsymbol{x}, \boldsymbol{y})$ can be decomposed into image regions $\boldsymbol{x}^{(1)}, \ldots, \boldsymbol{x}^{(m)}$, and there exist fine-grained captions $\boldsymbol{y}'^{(m)}$ that describe the corresponding image regions $\boldsymbol{x}^{(m)}$. Thus, the original input $(\boldsymbol{x}, \boldsymbol{y})$ becomes $\{(\boldsymbol{x}^{(1)}, \boldsymbol{y}'^{(1)}), \ldots, (\boldsymbol{x}^{(m)}, \boldsymbol{y}'^{(m)})\}$ for region-text considerations, and $(\boldsymbol{x}, \boldsymbol{y})$ is a special case when the "region" itself is the whole image. Based on this, we identify several research questions and will answer them in the following sections:

1. Considering the goal is to train an image encoder $f_I$ with enhanced localization capability, how should we formulate a region-text alignment goal that improves $f_I$? We propose a novel learning task called *promptable embeddings* in Section 3.2.
2. How to properly extract region embedding from $f_I(\boldsymbol{x})$ as an effective joint design? We propose a lightweight promptable region extractor in Section 3.3.
3. How to generate meaningful image regions with high-quality captions? Furthermore, in many cases, the ideal region caption $\boldsymbol{y}'^{(m)}$ may not exist in the image-level caption, *i.e.*, $\boldsymbol{y}'^{(m)}$ might

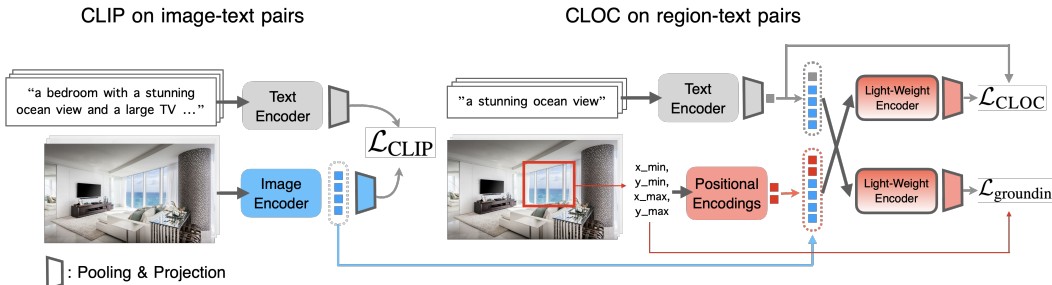

Figure 2: **CLOC promptable embedding architecture.** CLOC builds upon the image embedding from CLIP (before pooling and projection) and transforms it into a region-aware vision embedding given an encoded prompt; *e.g.*, positional encodings of box coordinates or regional caption encoded by the CLIP text encoder.

not be a substring of the original $\boldsymbol{y}$. We design an effective and scalable data engine as a visually-enriched and spatially-localized labeler to generate high-quality region-text pairs in Section 4.

4. With the above considerations, we discuss how to train the model with minimal conflicts towards a drop-in replacement of the CLIP model in Section 3.4.

### 3.2 PROMPTABLE EMBEDDINGS

To optimize CLIP with better feature localization and eventually learn an enhanced CLIP vision encoder $f_I$ for various VL downstream tasks, we argue that it will require at least two capabilities. ($i$) First, the encoder should recognize fine-grained small objects (*e.g.*, this image crop is an "airplane wheel"). ($ii$) Second, the *image embedding* produced by the encoder provides a holistic understanding such that an MLLM can reason more advanced spatial hierarchy relationships within the scene (*e.g.*, "The plane is lowering its front landing gear."). As discussed in Section 2, many previous works improve CLIP toward object detection tasks thus mainly focusing on ($i$) only; *e.g.*, RegionCLIP (Zhong et al., 2022) that crops out image regions and uses them as additional input images to re-train the CLIP encoders for recognizing objects. However, to support comprehensive VL tasks, ($i$) is necessary but insufficient without ($ii$).

To achieve this, we introduce a new concept, *promptable embedding*. We consider a scenario similar to MLLM use cases, where answers are generated using CLIP image tokens alongside a question. We hypothesize that a strong encoder for MLLMs should produce an *image* embedding that can *easily be transformed into region representations, given location cues*.

We re-formulate the CLIP loss based on image-text pairs $(\boldsymbol{x}, \boldsymbol{y})$ into a **localized** language-image contrastive loss for region-text alignment based on triplets of $(\{\boldsymbol{l}\}, \boldsymbol{x}, \boldsymbol{y})$, where $\boldsymbol{l}$ is a location representation such as a bounding box, and possibly there are several boxes as a set $\{\boldsymbol{l}\}$ per image. To make it compatible with CLIP training, we construct a promptable embedding transform module, or in short, *region prompter* $\boldsymbol{z} = \texttt{Prompter}(\boldsymbol{l}, f_I(\boldsymbol{x}))$, that extracts the region embedding specified by $\boldsymbol{l}$ from the image embedding $f_I(\boldsymbol{x})$. This formulation is inspired by the success of the segmentation model SAM (Kirillov et al., 2023) that predicts the *segmentation masks* conditioned on location prompt (*e.g.*, a box), while CLOC predicts a *region embedding* conditioned on $\boldsymbol{l}$ instead.

To this end, we decompose the location-image-text triplets as localized region-text pairs. Let $\boldsymbol{z}_i^{(m)} = \texttt{Prompter}(\boldsymbol{l}_i^{(m)}, f_I(\boldsymbol{x}_i))$ and $\boldsymbol{y}_i^{(m)}$ is the caption of the region specified by $\boldsymbol{l}_i^{(m)}$. $\boldsymbol{l}_i^{(m)} \in \mathbb{R}^4$ is the $m$-th box of image $i$ represented as two coordinates (*i.e.*, top-left and bottom-right corners). We then formulate a symmetric region-text contrastive loss similar to Equation 1:

$$\mathcal{L}_{R \to T} := -\frac{1}{MN} \sum_{i=1}^{N} \sum_{\boldsymbol{l}_i^{(m)} \sim \{\boldsymbol{l}_i\}} \log \frac{\exp\left(\texttt{sim}\left(\boldsymbol{z}_i^{(m)}, f_T(\boldsymbol{y}_i^{(m)})\right)/\tau\right)}{\sum_{j=1}^{N} \sum_{\boldsymbol{l}_j^{(m)} \sim \{\boldsymbol{l}_j\}} \exp\left(\texttt{sim}\left(\boldsymbol{z}_i^{(m)}, f_T(\boldsymbol{y}_j^{(m')})\right)/\tau\right)}, \quad (2)$$

where $M$ is the number of regions $\boldsymbol{l}_i^{(m)}$ sampled per image. We set $M = 4$ by default. We will discuss implementing the $\texttt{Prompter}$ in Section 3.3, and generating $\boldsymbol{l}_i^{(m)}$ with $\boldsymbol{y}_i^{(m)}$ in Section 4.

$\mathcal{L}_{T \to R}$ is the symmetric contrastive loss normalized along text-to-region axis, just like in Equation 1. We define $\mathcal{L}_{\text{CLOC}} = (\mathcal{L}_{R \to T} + \mathcal{L}_{T \to R})/2$.

As the `Prompter` is a simple transformer encoder, it allows flexible types of prompts besides bounding boxes we have used, such as points, free-form referring, text, and etc. We further consider the case where the prompt is free-form text, and leave others for future study. We add a grounding loss that extracts a region feature from the image (*e.g.*, a picture of the bedroom) given its regional caption (*e.g.*, "a large TV"), and predicts the bounding box with an MLP regression head, *i.e.*,

$$\mathcal{L}_{\text{grounding}} := \frac{1}{4MN} \sum_{i=1}^{N} \sum_{\boldsymbol{l}_i^{(m)} \sim \{\boldsymbol{l}_i\}} \|\boldsymbol{l}_i^{(m)} - \texttt{BoxHead}\big(\boldsymbol{z}(\boldsymbol{y}_i^{(m)})\big)\|_2, \qquad (3)$$

where $\boldsymbol{z}(\boldsymbol{y}) := \texttt{Prompter}(f_T(\boldsymbol{y}), f_I(\boldsymbol{x}))$ is the grounded embedding conditioned on the text (encoded by the CLIP text encoder). The overall loss is

$$\mathcal{L} := \mathcal{L}_{\text{CLIP}} + \lambda(\mathcal{L}_{\text{CLOC}} + \mathcal{L}_{\text{grounding}}), \qquad (4)$$

where $\lambda$ is a weighting scalar. In experiments, we set $\lambda$ to be the ratio of images in the mini-batch that contain region labels without extra tuning. All the learnable parameters are trained end-to-end.

## 3.3 CLOC Model Architecture

We implement the promptable embedding introduced in Section 3.2 with minimal extra modules on top of the original CLIP image and text encoders. As illustrated in Figure 2, the original CLIP model remains the same for computing $\mathcal{L}_{\text{CLIP}}$. For computing $\mathcal{L}_{\text{CLOC}}/\mathcal{L}_{\text{grounding}}$, the image embedding is reused from the CLIP ViT but before the pooled projection and normalization $f_I'$. To extract the region embedding $\boldsymbol{z} = \texttt{Prompter}(\boldsymbol{l}, f_I'(\boldsymbol{x}))$ from the image, we consider the location representation $\boldsymbol{l}$ as two coordinates (top-left and bottom-right corners of a box), each vectorized by positional encoding. The `Prompter` is a simple and lightweight one-layer transformer encoder. It takes the positional encodings prepended with the sequence of image tokens from ViT together as the input, and outputs the region embedding with a pooled projection layer. For the grounding loss, we reuse the same CLIP text encoder for encoding the region captions $\boldsymbol{z} = \texttt{Prompter}(f_T(\boldsymbol{y}), f_I'(\boldsymbol{x}))$ to predict the bounding boxes with a two-layer MLP head. Overall, CLOC only adds additional learnable parameters of the lightweight `Prompter`. Note, that the main overheads in a single forward are from encoding the image via the ViT – CLOC reuses it for multiple prompts.

## 3.4 Discussions on Design Choices and Extensions

We provide discussions here on the rationale behind our design choices and some minor extensions.

**Extracting region embedding with visual prompts.** To train our model with $\mathcal{L}_{\text{CLOC}}$ in Equation 4, it requires extracting region embeddings from the image features given the bounding boxes. A perhaps straightforward alternative could be Region-of-Interest (RoI) pooling/alignment (He et al., 2017) from the spatial image feature of ViT before pooling. RoI operations are popular, especially in the object detection literature.[1] However, as will be evidenced by worse performance in Section 5, we found it suboptimal for CLOC pre-training here for several reasons. First, unlike object detection datasets that typically contain golden labels, here the pseudo-labels are much noisier on the large-scale web-crawled images. Therefore, the resulting RoI features may be inaccurate due to the imprecise bounding boxes, making model training less effective. Second, unlike dense vision tasks that directly rely on the spatial features, MLLM has a transformer decoder that consists of several attention layers such that the constraint of semantics in the spatial feature space becomes somewhat indirect. Our `Prompter` mimics such inductive bias in pre-training via a single-attention-layer encoder that may leverage better global context reasoning compared to RoIs.

**Avoiding region-text conflicts.** While region annotations introduce location information, a concern of contrastive learning on the regional captions may be that there are many similar objects within an image (*e.g.*, "boats" in the harbor) or a mini-batch. To mitigate such concerns, we apply two tricks. First, fortunately, we found it sufficient to sample a few regions per image for each update, *e.g.*, we

---

[1] We observe a withdrawn arXiv preprint (https://arxiv.org/abs/2401.06397) proposes to extract RoI features for CLIP regional contrastive learning but only focus on dense vision tasks, not MLLMs.

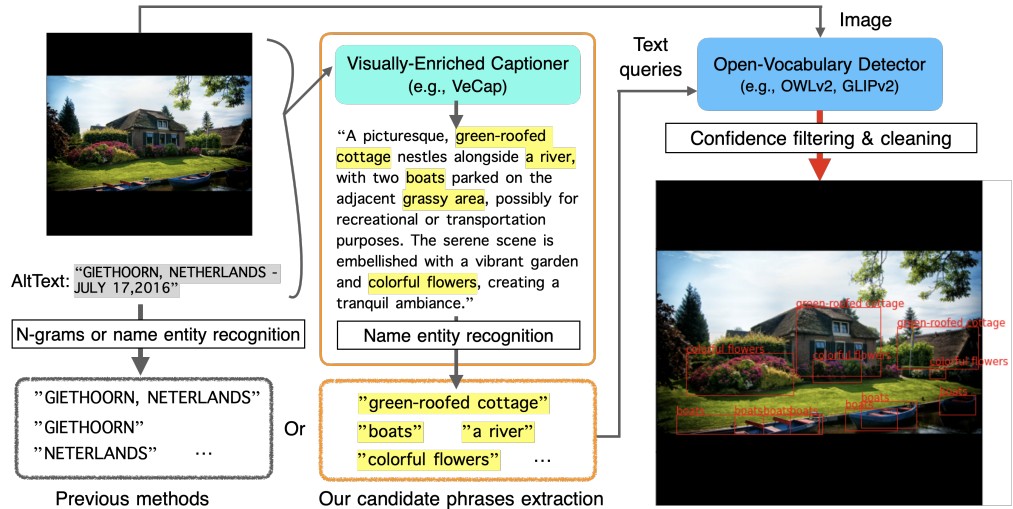

Figure 3: **Overview of the Visually-Enriched and Spatially-Localized (VESL) captioning pipeline.** We leverage an existing open-vocabulary detector (*e.g.*, OWLv2) that typically predicts bounding boxes on the images, and assigns the labels from the given text phrase candidates. Previous methods do not tailor how the text phrases are prepared and often use the alt-text attached to the images, which is prone to insufficient region descriptions. We found it crucial for CLOC to train on data from our VESL that re-captions images with the visually-enriched captioner VeCap (Lai et al., 2024) for better visual concept exploitation of the detector.

Table 1: **Region-text dataset statistics.** We summarize the text token length for both images and regions. Partial statistics of the proprietary datasets revealed by their papers. *The 20M subset of GRIT is released at: `https://huggingface.co/datasets/zzliang/GRIT`; we removed the invalid images.

| Dataset | # of images | regions per image | image caption length | region text length |
|---|---|---|---|---|
| Flickr Entities (Plummer et al., 2015) | 32K | 8.7 | – | – |
| RefCOCO (Yu et al., 2016) | 20K | 2.5 | – | 3.6 |
| RefCOCO+ (Yu et al., 2016) | 20K | 2.5 | – | 3.5 |
| RefCOCOg (Mao et al., 2016) | 27K | 2.1 | – | 8.4 |
| Visual Genome (Krishna et al., 2017) | 108K | 38.0 | – | – |
| GRIT (proprietary) (Peng et al., 2023) | 91M | 1.5 | – | 4.7 |
| GRIT (released, clean) (Peng et al., 2023)* | 17M | 1.8 | 17.2 | 4.6 |
| Florence-2 (proprietary) (Xiao et al., 2024) | 126M | 5.4 | 70.5 | 2.6 |
| OWLv2 (proprietary) (Minderer et al., 2024) | 2B | – | – | – |
| WiT labeled w/ Minderer et al. (2024) | 300M | 5.1 | 17.1 | 3.9 |
| VESL WiT (Ours) | 300M | 11.6 | 44.9 | 2.1 |
| VESL WiT+DFN (Ours) | 2B | 11.5 | 35.9 | 2.1 |

set $M = 4$ in Equation 2 in experiments. Second, we can filter similar texts when computing the negatives in the contrastive loss. More specifically, we ignore the pairs of $\left(\boldsymbol{z}_i^{(m)}, f_T(\boldsymbol{y}_j^{(m')})\right)$ in the denominators of both $\mathcal{L}_{R \to T/T \to R}$, if $\texttt{sim}\left(f_T(\boldsymbol{y}_i^{(m)}), f_T(\boldsymbol{y}_j^{(m')})\right) > 0.9$, without gradients on $f_T$.

## 4 VISUALLY-ENRICHED AND SPATIALLY-LOCALIZED CAPTIONING PIPELINE

As discussed in Section 1 and 3.1, a key bottleneck of CLOC training is the region-text annotation datasets in terms of both the data size scales and the label quality, since there are no public datasets with region-text annotations at scales large enough for contrastive pre-training.

Inspired by recent works that re-caption images with visually-enriched captions for better CLIP training, we make a step further for **V**isually-**E**nriched and **S**patially-**L**ocalized (**VESL**) labeler which generates more fine-grained captions at the region level. The goal of VESL is, given an

image (possibly with the original web-crawled alt-text), annotate it with the grounded bounding boxes each associated with a caption in natural language for optimizing Equation 2 in Section 3.2.

Concretely, VESL is constructed as a pseudo-labeling pipeline with the following steps:

1. **Image re-captioning with visual concept exploitation**: We follow the VeCap framework (Lai et al., 2024) to generate long, diverse, and detailed image captions.
2. **Region phrase candidates extraction**: Inspired by Zhang et al. (2022), we apply name entity recognition (NER) to extract leaf entities from the visually-enriched captions as potential candidate phrases describing a region inside the image.
3. **Open-vocabulary detection with extracted phrases**: we generate the final region-text annotations via a pre-trained open-vocabulary detector queried with the phrases extracted from Step 2 to match the bounding boxes proposed by the detector. We adopted the OWLv2 detector (Minderer et al., 2024) which contains the CLIP image/text encoder with the detection head. The boxes with detection confidence larger than $0.1$ are kept as the region location and the phrases matched with the highest score are considered as their captions.

**Remarks.** We highlight our insights behind the proposed recipe. The most relevant work was proposed in (Minderer et al., 2024) that scales up open-vocabulary (OV) detection via self-training. We are inspired by its success and extend it to CLOC contrastive learning with important modifications. Different from (Minderer et al., 2024) that generates candidate phrases from the $n$-grams of the web-crawled alt-text of the images for OV detection, we found the alt-text might not have enough details describing the image region content, thus limiting the diversity and quality of the annotations predicted by the OV detector. We thus caption each image augmented with more visual details. However, the long captions make the $n$-grams candidates verbose and grow exponentially, thus we generate high-quality candidates via name entity recognition instead. We found such a pipeline produces training data more suitable for CLOC, as will be validated in Section 5.

**Our pre-training datasets.** Our pre-training data consists of two parts: ($i$) image-text pairs, and ($ii$) region-text pairs. For image-text pairs, we reproduce the image re-captioning pipeline from VeCap (Lai et al., 2024), and generate synthetic captions for WiT-300M (Wu et al., 2024) and DFN-5B (Fang et al., 2023) images. For region-text pairs, we pseudo-label WiT-300M and a 2B-image subset of DFN-5B using our VESL pipeline. In VESL, we adopted the official OWLv2 L/14 model (Minderer et al., 2024) as the open-vocabulary detector.[2] All images are pseudo-labeled with $448 \times 448$ resolution, where a maximum number of 20 phrase queries are sampled for moderate computation budget. Table 1 summarizes the statistics of existing region-text datasets and ours. Notably, we also ablate annotating WiT-300M following Minderer et al. (2024) and found it detects *less* objects with longer region text, likely due to verbose $n$-grams of alt-text are in lower quality than our approach, as discussed in the remarks. Examples and pseudo codes are in Appendix B.

## 5 EXPERIMENTS

### 5.1 SETUP

**Pre-training.** We follow OpenAI-CLIP (Radford et al., 2021) to train both our CLIP baseline model and CLOC model using a similar budget of around 14B images seen[3]. For a fair comparison, we use the same hyper-parameters and images for both the CLIP baseline and CLOC. We experimented with the ViT B/16 and L/14 architectures, pre-trained with $224 \times 224$ and $336 \times 336$ image resolutions, respectively. All parameters are trained end-to-end from scratch. We implement the codebase in JAX (Bradbury et al., 2018). We provide hyper-parameters and more details in Appendix A.

**Evaluation tasks.** We evaluate our image encoders across a wide range of downstream tasks. First, we assess performance on ImageNet image classification (Deng et al., 2009; Shankar et al., 2020) and COCO retrieval (Lin et al., 2014). Second, we construct region-level tasks, including COCO object recognition and region-text retrieval using the GRIT dataset (Peng et al., 2023). Furthermore, we show CLOC is particularly useful for MLLMs, validated by the Ferret model (You et al., 2023) which requires fine-grained image understanding for referring and grounding tasks. We also evaluate on general multimodal benchmarks using LLaVA-1.5 (Liu et al., 2023) and LLaVA-NeXT (Liu

---

[2]OWLv2 CLIP L/14 ST+FT in: https://github.com/google-research/scenic/tree/main/scenic/projects/owl_vit

[3]To avoid confusion, we will refer to CLIP as training with the standard CLIP loss on the same data as CLOC and refer to OpenAI-CLIP as the public pre-trained checkpoint released from Radford et al. (2021).

Table 2: Zero-shot evaluation on image-level tasks (accuracy of ImageNet (IN) classification, recall@1 of COCO retrieval) and region-level tasks (mAcc of region object recognition on COCO and LVIS, recall@10 of GRIT region retrieval), using ViT-B/16 as the default encoder backbone. The indentation with different symbols denotes removing (–) or changing a component (∘).

| Models | Training data | | Image tasks | | | | Region tasks | | | | Avg. | |
|---|---|---|---|---|---|---|---|---|---|---|---|---|
| | Image | Region | COCO (i2t) | COCO (t2i) | INv1 | INv2 | GRIT (r2t) | GRIT (t2r) | COCO Recog. | LVIS Recog. | Image | Region |
| ① OpenAI-CLIP | proprietary | - | 52.4 | 33.1 | 68.3 | 62.3 | - | - | - | - | 54.0 | - |
| ② CLIP | WiT+DFN | - | 66.3 | 45.1 | 76.2 | 69.6 | - | - | - | - | 64.3 | - |
| ③ CLOC | WiT | WiT | 68.8 | 50.1 | 66.7 | 59.7 | 65.1 | 67.2 | 70.6 | 26.7 | 61.3 | 57.4 |
| ④ − Prompter | WiT | WiT | 67.0 | 49.7 | 65.6 | 58.6 | 44.8 | 4.4 | 55.3 | 13.2 | 60.2 | 29.4 |
| ⑤ − VESL | WiT | WiT | 53.9 | 36.3 | 66.6 | 59.5 | 71.5 | 63.8 | 62.2 | 22.2 | 54.1 | 54.9 |
| ⑥ ∘ w/ GLIPv2 | WiT | WiT | 68.8 | 50.0 | 65.8 | 59.2 | 67.9 | 71.1 | 64.9 | 23.1 | 61.0 | 56.8 |
| ⑧ CLOC | WiT+DFN | WiT | 66.1 | 46.5 | 75.5 | 68.6 | 65.8 | 67.4 | 70.1 | 27.2 | 64.2 | 57.6 |
| ⑨ − Prompter | WiT+DFN | WiT | 65.8 | 46.5 | 75.7 | 68.0 | 55.5 | 18.4 | 67.1 | 24.6 | 64.0 | 41.4 |
| ⑩ − text filtering | WiT+DFN | WiT | 65.4 | 46.0 | 75.7 | 68.4 | 66.3 | 66.5 | 68.7 | 24.8 | 63.9 | 56.6 |
| ⑪ − $\mathcal{L}_{\text{grounding}}$ | WiT+DFN | WiT | 66.0 | 46.3 | 75.7 | 67.9 | 66.0 | 66.8 | 70.0 | 25.8 | 64.0 | 57.2 |
| ⑫ ∘ $M=2$ | WiT+DFN | WiT | 66.6 | 46.2 | 75.5 | 67.9 | 66.5 | 67.0 | 69.8 | 25.8 | 64.1 | 57.3 |
| ⑬ CLOC | WiT+DFN | WiT+DFN | 69.2 | 49.3 | 74.9 | 67.0 | 63.9 | 65.9 | 71.1 | 28.5 | 65.1 | 57.3 |
| ⑭ − Prompter | WiT+DFN | WiT+DFN | 70.2 | 49.7 | 74.7 | 67.6 | 65.7 | 23.0 | 67.1 | 25.4 | 65.6 | 45.3 |
| ⑮ − VESL | WiT+DFN | WiT+DFN | 65.3 | 46.6 | 75.5 | 67.7 | 55.7 | 22.3 | 66.3 | 25.3 | 63.8 | 42.4 |
| ⑯ ∘ ViT L/14 | WiT+DFN | WiT+DFN | 74.8 | 54.4 | 80.1 | 73.2 | 66.9 | 68.3 | 72.9 | 32.6 | 70.6 | 60.2 |
| ⑰ ∘ ViT H/14 | WiT+DFN | WiT+DFN | 75.7 | 55.1 | 81.3 | 74.7 | 67.4 | 69.4 | 73.0 | 35.6 | 71.7 | 61.3 |

et al., 2024)[4], which both use the 7B Vicuna LLM. For all evaluation tasks, we use the same official hyper-parameters, fine-tuning datasets, and codebase for all the image encoders we experimented with, without specific tuning. More details are provided in each subsection and in Appendix A.

## 5.2 Image and Region Classification and Retrieval Tasks

The proposed CLOC training framework enables the encoder to produce not only image embedding but also region embeddings. It can directly be used for region-level tasks without further training, in analogy to the zero-shot capability of CLIP on images. To evaluate such capability and also for fast development and ablation study, we first construct several region-level zero-shot tasks.

Besides *image*-level evaluation like ImageNet classification and COCO image-text retrieval, we additionally construct *region*-level tasks, including region object recognition and region-text retrieval. More specifically, the region-level tasks leverage the labeled bounding boxes in the evaluation set for CLOC to extract region embedding. For region retrieval, we use a validation set of the GRIT dataset (Peng et al., 2023) and encode both the image regions and the region captions. For region classification, the class names are encoded as text embedding (80 / 1203 classes for COCO / LVIS, respectively), and the highest cosine similarity for each region embedding is predicted as its class.

We highlight important variables for the performance in Table 2 with the following observations:

- CLOC performs decently on region-level tasks[5] while maintaining strong performance on image-level metrics (② vs. ⑧ ⑬).
- The Prompter is an important ingredient for CLOC's success to go beyond CLIP without a compromise (③ vs. ④; ⑧ vs. ⑨; ⑬ vs. ⑭). We replace the Prompter with RoI alignment to extract region features and train with $\mathcal{L}_{\text{CLOC}}$. We found it performs much worse on region-level

---

[4]We use the codebase: https://github.com/xiaoachen98/Open-LLaVA-NeXT.

[5]For reference, in a different data setup, Wu et al. (2023) reports 46.5% mAcc on the same COCO region classification task, trained using $320 \times 320$ COCO training images. In contrast, our approach achieves over 70% mAcc, pre-trained on a $224 \times 224$ large-scale web-crawled dataset with object annotations (thus might not be fair comparisons).

Table 3: Results on Ferret-Bench for referring and grounding VQA, based on Ferret (You et al., 2023) equipped with different image encoders. Models are evaluated with OpenAI `gpt-4o` API instead of the deprecated `gpt-4-0314` in the paper. *replace Ferret visual sampler with `Prompter`; see Section 5.3 for details.

| Method | ViT | Region Alignment | # of images w/ region labels | Referring Description | Referring Reasoning | Grounding in Conversation | Avg. ($\Delta$ to CLIP) |
|---|---|---|---|---|---|---|---|
| CLIP | B/16 | None | None | 47.5 | 50.3 | 45.3 | 47.7 |
| CLOC | B/16 | RoI-Align | 300M | 48.0 | 48.4 | 40.0 | 45.5 |
| CLOC | B/16 | Prompter | 300M | 50.2 | 55.5 | 41.5 | 49.1 |
| CLOC | B/16 | Prompter | 2B | 53.6 | 53.7 | 42.2 | 49.8 (+2.1) |
| CLOC * | B/16 | Prompter | 2B | 54.8 | 54.9 | 44.7 | **51.5** (+3.7) |
| OpenAI-CLIP | L/14 | None | None | 50.8 | 55.4 | 45.7 | 50.6 |
| CLIP | L/14 | None | None | 54.2 | 54.6 | 43.3 | 50.7 |
| CLOC | L/14 | Prompter | 300M | 51.0 | 65.7 | 44.9 | 53.9 |
| CLOC | L/14 | Prompter | 2B | 55.9 | 63.3 | 46.0 | 55.1 (+4.4) |
| CLOC * | L/14 | Prompter | 2B | 56.3 | 67.4 | 47.1 | **56.9** (+6.2) |

tasks than CLOC, possibly due to such strong constraints of RoI features being difficult to learn on the noisy labels with the CLIP loss as discussed in Section 3.4.

- VESL outperforms Minderer et al. (2024) baseline approach, as the visually-enriched captions improve image retrieval tasks (as expected (Lai et al., 2024)), while also offering versatile visual concepts as text candidates for the OV detector, supporting Section 4 (③ vs. ⑤; ⑬ vs. ⑮).
- Given the same captions in VESL, OWLv2 slightly outperforms the GLIPv2 detector (③ vs. ⑥).
- Tricks in Section 3.4 offer slight performance gains, but $\mathcal{L}_{\text{CLOC}}$ is already highly effective on its own (⑩ ⑪).
- Region tasks work well when sampling 2 or 4 boxes per image, making CLOC practical (⑫).
- Scaling up region labels seems saturated at 300M images on region tasks (③ ⑧ ⑬), while we found it will further improve in MLLM tasks as will be shown in Table 3.
- Scaling up the ViT model sizes can further improve both image and region tasks (⑬ ⑯ ⑰).

Overall, CLOC not only achieves strong performance on image-level tasks, but unlocks a new capability for zero-shot region-level tasks. We have validated our design choices for architectures, training, and data. Below, the complete setup ⑬ will be used as default if not specified.

## 5.3 REFERRING AND GROUNDING WITH FERRET

As discussed in Section 1, a key motivation is to provide an enhanced image encoder for training MLLMs, particularly for tasks requiring fine-grained image understanding. A notable example is Ferret (You et al., 2023), a recently proposed MLLM that builds on LLaVA and aims to handle more advanced spatial interactions, such as referring and grounding in VQA tasks. Ferret can take region prompts such as a box, a point, or a free-form location referring to the input image as input, and answer a question specific to the region such as "Do you know when the object[region] was invented?" Ferret thus requires fine-grained image features from the vision encoder for spatial reasoning.

We evaluate CLOC by replacing the CLIP ViT encoder with our CLOC ViT as a drop-in replacement. We follow the official codebase[6] for training the Ferret model. We further consider a variant based on Ferret: the Ferret model implements a spatial-aware visual sampler that samples image features from the region specified in the question. We replace the sophisticated visual sampler with our simple `Prompter` introduced in Section 3.3 to extract region embedding with $z = \text{Prompter}(l, f'_I(x))$ instead, as illustrated in Figure 1(right).

In Table 3, we evaluate different pre-trained image encoders on the Ferret-Bench benchmark (You et al., 2023). Ferret-Bench includes challenging multimodal dialogue-style VQA of three tasks constructed with GPT-4. Results show that our `Prompter` is essential to improve upon the CLIP baseline – RoI-Align may even slightly degrade performance. Scaling region labels from 300M to 2B further improves performance. Interestingly, our `Prompter` (denoted as *) can be a replacement of the FERRET visual sampler in fine-tuning, which is simpler and performs even better up to 6%

---

[6]We use the official Ferret codebase: `https://github.com/apple/ml-ferret`.

Table 4: Results on referred object classification (LVIS), referring expression comprehension (0.5 IoU on RefCOCO, RefCOCO+, RefCOCOg), and phrase grounding (0.5 IoU on Flickr30k Entities) with Ferret. Shikra: baseline in Chen et al. (2023). Ferret*: replace visual sampler with CLOC prompter.

| Model | Encoder | LVIS | | | RefCOCO | | | RefCOCO+ | | | RefCOCOg | | Flickr | | Avg. |
| | | box | point | free-form | val | testA | testB | val | testA | testB | val | test | val | test | ($\Delta$ to CLIP) |
|---|---|---|---|---|---|---|---|---|---|---|---|---|---|---|---|
| FERRET | CLIP B/16 | 72.5 | 56.9 | 57.2 | 80.7 | 84.2 | 77.1 | 71.9 | 76.1 | 63.7 | 75.9 | 76.2 | 76.2 | 78.3 | 72.8 |
| FERRET | CLOC B/16 | 74.3 | 56.7 | 60.2 | 84.2 | 87.0 | 80.0 | 74.7 | 80.0 | 67.0 | 78.8 | 79.5 | 80.0 | 81.5 | 75.7 (+2.9) |
| FERRET * | CLOC B/16 | 78.9 | 58.2 | 61.4 | 84.4 | 86.8 | 78.9 | 74.0 | 78.7 | 65.5 | 78.0 | 78.7 | 80.1 | 81.4 | **75.8** (+3.0) |
| Shikra | OpenAI-CLIP L/14 | 57.8 | 67.7 | n/a | 87.0 | 90.6 | 80.2 | 81.6 | 87.4 | 72.1 | 82.3 | 82.2 | 75.8 | 76.5 | - |
| FERRET | OpenAI-CLIP L/14 | 79.4 | 67.9 | 69.8 | 87.5 | 91.4 | 82.5 | 80.8 | 87.4 | 73.1 | 83.9 | 84.8 | 80.4 | 82.2 | 80.8 |
| FERRET | CLIP L/14 | 78.7 | 66.9 | 70.2 | 88.0 | 90.4 | 83.5 | 80.1 | 85.8 | 73.3 | 82.8 | 83.4 | 79.0 | 80.1 | 80.2 |
| FERRET | CLOC L/14 | 81.6 | 67.9 | 69.9 | 89.0 | 91.0 | 84.7 | 81.4 | 86.8 | 74.7 | 84.0 | 85.2 | 82.3 | 83.3 | **81.7** (+1.5) |
| FERRET * | CLOC L/14 | 79.8 | 67.9 | 69.1 | 88.2 | 91.1 | 84.5 | 80.6 | 86.7 | 73.9 | 84.8 | 85.1 | 82.4 | 83.5 | 81.4 (+1.2) |

Table 5: Results on multimodal benchmarks using LLaVA-1.5/NeXT with ViT-L/14 and Vicuna-7B.

| Method | MLLM | LLaVA$^W$ | TextVQA | GQA | MM-Vet | POPE | MME-P | MME-C |
|---|---|---|---|---|---|---|---|---|
| CLIP | LLaVA-1.5 | 59.3 | 53.3 | 62.2 | 30.0 | 86.7 | 1451.4 | 254.3 |
| CLOC | LLaVA-1.5 | 64.3 | 54.9 | 62.7 | 31.5 | 87.3 | 1482.0 | 288.9 |
| CLIP | Open-LLaVA-NeXT | 67.3 | 61.4 | 63.5 | 38.5 | 87.9 | 1486.1 | 279.6 |
| CLOC | Open-LLaVA-NeXT | 69.5 | 61.9 | 64.2 | 40.2 | 88.3 | 1451.1 | 312.5 |

against both the OpenAI-CLIP and our in-house CLIP. We also evaluate CLOC (2B labeled) on other referring and grounding tasks ranging from referring object classification, referring expression comprehension, and phrase grounding across multiple datasets. As summarized in Table 4, CLOC is also superior evidenced by $1 \sim 3\%$ improvements in average of 13 evaluation sets.

## 5.4 GENERAL VQA WITH LLAVA-1.5 AND LLAVA-NEXT

We further show that the CLOC encoder is also competitive against CLIP on general VQA tasks without regression and can even provide performance improvements. We use the Vicuna 7B LLM decoder for two experiments based on LLaVA-1.5 (frozen encoder) and Open-LLaVA-NeXT (unfrozen encoder with AnyRes (Liu et al., 2024) inputs). Since general VQA does not provide spatial referring inputs, we simply replace the ViT in LLaVA. Table 5 summarizes the results. Encouragingly, with our CLOC designs, the improved region-level alignment is also beneficial to some general multimodal benchmarks, as they may also require fine-grained image understanding.

## 6 CONCLUSION

Please see Appendix C for more discussions where we comment on the limitations, future directions, computation cost, design rationales, etc.

We tackle a deficiency of CLIP, to make the semantics aligned in the vision space not only at the image level but also at the region level. We propose a new pre-training framework that includes innovations in a new learning formulation that a strong encoder should be easily transformed in the foresee of downstream use of MLLMs. Our encoder creates a new possibility for adapting the features with input prompts of interaction together with MLLMs. To resolve the need for large-scale region-text training data, we carefully design a pseudo-labeling pipeline for visually-enriched and spatially-localized captions. Our pre-trained encoder is essentially a drop-in replacement of CLIP, with competitive image-text performance, and extra capability demonstrated in region-text tasks and VQA tasks with MLLMs.

## REPRODUCIBILITY STATEMENT

We made our best efforts to exhaustively state the implementation details. Training hyper-parameters and model architectures are discussed in Section 3.2, 3.3, and 5.1, with a summary in Appendix A and Table A. For evaluation, as mentioned in Section 5.1, we strictly follow the official setup with the codebase released by the original authors if applicable, with details provided in Section A.2. For our datasets, we provide data processing details in Section 4 and example codes in Appendix B. We are working hard on releasing the annotations with internal approvals.

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

# APPENDIX

## A  EXPERIMENT DETAILS

We provide the omitted experiment details for pre-training and the downstream evaluation tasks.

### A.1  PRE-TRAINING HYPER-PARAMETERS

For pre-training both the in-house CLIP baseline and CLOC, we mainly follow the hyper-parameters in (Radford et al., 2021) to train on our in-house datasets. The training images are identical for CLIP and CLOC, while CLOC is trained on the extra region-text annotations of the same images via the proposed VESL pipeline (details in Section B). Table A summarizes the general training hyper-parameters used for all experiments and the setup for components specific to CLOC.

In terms of the CLOC architecture, as illustrated in Figure 2, the image and text encoders including the attention pooling and projection layers follow the same as OpenAI-CLIP (Radford et al., 2021). Our `Prompter` consists of a positional encoding matrix for bounding boxes, and a single-layer single-head transformer encoder with another set of the global average pooler and a projection layer to map the region embeddings into the same dimension as the CLIP text/image embeddings.

Table A: Pre-training hyper-parameters and settings for the in-house CLIP baseline and CLOC.

| General | |
|---|---|
| Batch size | 32768 |
| Image size | $224 \times 224$ (ViT B/16) or $336 \times 336$ (ViT L/14, H/14) |
| Image pre-processing | long-side resizing with padding (i.e., `tf.image.resize_with_pad`) |
| Text tokenizer | T5 (Raffel et al., 2020), lowercase |
| Text maximum length | 77 tokens |
| Steps | 439087 (i.e., $\sim$ 14B examples seen) |
| Optimizer | AdamW ($\beta_1 = 0.9, \beta_2 = 0.98$) |
| Peak learning rate (LR) | 0.0005 |
| LR schedule | cosine decays with linear warm-up (first 2k steps) |
| Weight decay | 0.2 |
| Dropout rate | 0.0 |
| **CLOC** | |
| # of sampled regions | maximum $M = 4$ per image |
| CLOC loss weight | $\lambda = 1.0 \times \frac{\text{\# of images contain region text in the mini-batch}}{\text{batch size}}$ (in Equation 4) |
| Encoding box prompts | sinusoidal positional encoding of coordinates (top-left and bottom right of a box) |
| Encoding text prompts | encoded by re-using the text encoder (w/ pooling & projection) |
| `Prompter` architecture | a single-layer single-head transformer encoder (same feature dimension as the ViT) |
| `BoxHead` architecture | 2-layer MLP with GELUs activations (Hendrycks & Gimpel, 2016) |

### A.2  EVALUATION TASKS

We provide more details about the tasks constructed for evaluating the encoders in Section 5.

**Zero-shot region tasks.**  Our CLOC training augments a new capability for CLIP to generate region-level embeddings. This enables us to perform zero-shot region-text tasks, in analogy to the image-text zero-shot tasks like ImageNet classification and COCO text-image retrieval that CLIP has been evaluated on.

In a similar rationale of image-level evaluation, we further construct region-level tasks including region object recognition and region-text retrieval. For region object recognition, the class names are encoded by the text encoder into class embedding. We do not add the text prompts (e.g., "a photo of ...") to object classes used when CLIP (Radford et al., 2021) evaluated on image classification. The CLOC model takes all the labeled bounding boxes in the images to generate a region embedding

$\boldsymbol{z} = \texttt{Prompter}(\boldsymbol{l}, f_I(\boldsymbol{x}))$. The class embedding with the highest similarity is predicted as the class of the region (i.e., out of $80$ / $1203$ classes for COCO / LVIS).

For region retrieval, similarly, the CLOC model encodes both the image regions and the region captions from the public region-text GRIT dataset that the regions are annotated by the Kosmos-2 pipeline (Peng et al., 2023). We randomly sampled a 2K image validation set for fast evaluation. We have verified it is statistically stable compared to the whole set that contains about 20M in total. Unlike image-text retrieval the image captions are likely unique, the objects in regions of many images might be duplicated. Therefore, we opt to report recall@10 rather than recall@1 for GRIT region retrieval in Table 2.

**MLLM tasks.** To demonstrate our CLOC can benefit MLLM end tasks as a better image backbone, we consider two sets of MLLM experiments.

First, we experiment with the FERRET MLLM that is capable of taking spatial referring inputs for grounding and referring VQA tasks. FERRET can consume a point, a bounding box, or a free-form referring. It designs a quite complicated visual sampler module that involves point sampling and $k$NN grouping. We suggest the readers refer to Figure 3 and Section 3.2 in (You et al., 2023) for more details. Here we consider two variants of use cases of CLOC compatible with FERRET: (1) we only take the ViT encoder in CLOC to replace the CLIP ViT and still use the original FERRET visual sampler or (2) we further replace the visual sampler with our simple `Prompter` (essentially a lightweight transformer encoder with box positional encodings) in Section 3.3 as illustrated in Figure 1(3b). More specifically, we simply convert all types of spatial referring as boxes. As evidenced by Table 3 and Table 4, our `Prompter` can indeed be a much simpler alternative and may perform even better as it is more consistent with CLOC pre-training.

Second, we evaluate on general VQA tasks that do not consider extra spatial referring inputs. The pre-trained ViT of CLOC is a drop-in-replacement of CLIP ViT in two sets of experiments of LLaVA-1.5 (Liu et al., 2023) and LLaVA-NeXT (Liu et al., 2024). The main difference includes different supervised fine-tuning (SFT) sets. Also, LLaVA-NeXT uses the AnyRes technique that decomposes an image into several subimages that are encoded independently with the ViT and concatenated together as the input for the decoder. LLaVA-1.5 by default freezes the ViT while LLaVA-NeXT fine-tunes all parameters during SFT. Since the official LLaVA-NeXT is trained on some proprietary datasets that are not reproducible, we use the Open-LLaVA-NeXT repository[7]. Our experiments in Table 5 demonstrate CLOC not only slightly improves such general VQA besides FERRET tasks but also generalizes well for both LLaVA-1.5 and LLaVA-NeXT settings.

# B  VESL DATA ENGINE

We provide more information about our pseudo-labeling data pipeline proposed in Section 4.

## B.1  IMPLEMENTATION DETAILS

As already mentioned in Section 4, there are three steps for VESL: image re-captioning, region phrase candidates extraction from the captions, and open-vocabulary (OV) detection given the region candidates as queries.

For the re-captioning, the goal is to replace AltText with long, diverse, and detailed captions that can be used to generate more visual concepts as the region candidate phrases for the OV detector. Technically, any strong image captioner can be an option. In our paper, we adopt the VeCap pipeline (Lai et al., 2024) and leverage their images with enriched captions.

To extract region phrase candidates from the long captions, we adopt name entity recognition (NER) to extract leaf entities from the captions, inspired by (Zhang et al., 2022). The code listing below shows the Python example implementation, where stop-words and common generic words are filtered, following (Minderer et al., 2024).

Generating bounding boxes and assigning region captions can be done by querying an OV objection detector. We adopted the OWLv2 detector (Minderer et al., 2024) with their pre-trained L/14 checkpoint to annotate inputs with $448 \times 448$ image resolutions.

---

[7]https://github.com/xiaoachen98/Open-LLaVA-NeXT

```python
from typing import Iterable, List
import nltk

# STOPWORDS_EN and COMMON_GENERIC_WORDS are following:
# Section A.2 (Minderer et al., 2024)

# Stopwords from nltk.corpus.stopwords.words("english"):
STOPWORDS_EN = frozenset({
    "a", "about", "above", "after", "again", "against", "all", "am", "an",
    "and", "any", "are", "as", "at", "be", "because", "been", "before", "being",
    "below", "between", "both", "but", "by", "can", "did", "do", "does",
    "doing", "don", "down", "during", "each", "few", "for", "from", "further",
    "had", "has", "have", "having", "he", "her", "here", "hers", "herself",
    "him", "himself", "his", "how", "i", "if", "in", "into", "is", "it", "its",
    "itself", "just", "me", "more", "most", "my", "myself", "no", "nor", "not",
    "now", "of", "off", "on", "once", "only", "or", "other", "our", "ours",
    "ourselves", "out", "over", "own", "s", "same", "she", "should", "so",
    "some", "such", "t", "than", "that", "the", "their", "theirs", "them",
    "themselves", "then", "there", "these", "they", "this", "those", "through",
    "to", "too", "under", "until", "up", "very", "was", "we", "were", "what",
    "when", "where", "which", "while", "who", "whom", "why", "will", "with",
    "you", "your", "yours", "yourself", "yourselves"
})

# These words were found by manually going through the most common 1000 words
# in a sample of alt-texts and selecting generic words without specific meaning:
COMMON_GENERIC_WORDS = frozenset({
    "alibaba", "aliexpress", "amazon", "available", "background", "blog", "buy",
    "co", "com", "description", "diy", "download", "facebook", "free", "gif",
    "hd", "ideas", "illustration", "illustrations", "image", "images", "img",
    "instagram", "jpg", "online", "org", "original", "page", "pdf", "photo",
    "photography", "photos", "picclick", "picture", "pictures", "png", "porn",
    "premium", "resolution", "royalty", "sale", "sex", "shutterstock", "stock",
    "svg", "thumbnail", "tumblr", "tumgir", "twitter", "uk", "uploaded", "vector",
    "vectors", "video", "videos", "wallpaper", "wallpapers", "wholesale", "www",
    "xxx", "youtube"
})

def _is_all_stopwords(query_words: Iterable[str]) -> bool:
    return set(query_words).issubset(STOPWORDS_EN)

def _get_name_entities(words: List[str]) -> List[str]:
    """
    Returns name entities of image caption as queries, similar to GLIP.
    """
    pos_tags = nltk.pos_tag(words)
    grammar = "NP: {<DT>?<JJ.*>*<NN.*>+}"
    cp = nltk.RegexpParser(grammar)
    result = cp.parse(pos_tags)

    queries = []
    for subtree in result.subtrees():
        if subtree.label() == "NP":
            query_words = [t[0] for t in subtree.leaves()]
            # Don't use it if it only consists of stop words.
            if _is_all_stopwords(query_words):
                continue
            queries.append(" ".join(query_words))
    return queries

def find_noun_phrases(
    caption: str, max_num_queries: int = 20,
) -> List[str]:
    caption = caption.lower()
    tokens = nltk.word_tokenize(caption)
    # Remove common generic words.
    words = [w for w in tokens if w not in COMMON_GENERIC_WORDS]
    queries = _get_name_entities(words)[:max_num_queries]
    return queries

candidate_quries = find_noun_phrases(caption)
```

Listing 1: Python example codes for Step 2 of VESL in Section 4 for extracting text candidate queries from a caption.

### B.2 MORE VISUALIZATIONS

As mentioned in the remarks of Section 4, we found the AltText sourced from the original web-crawled images might not have enough details describing the subimage content, thus limiting the diversity and quality of the text candidate queries for the OV detector to detect more meaningful objects. In Figure A we show some cherry-picked examples (since the web-crawled images are quite noisy) just to demonstrate the reasons why high-quality captions can help our region-text annotation pipeline. In (Minderer et al., 2024), the queries are generated by the $n$-grams of the AltText, while ours are by NER as described in Section B.1 on top of the visually-enriched re-captions. Note that, in both methods we use the same pre-trained OV detector but with different approaches to generate the queries.

As shown in Figure A, for easier images like the first row, both methods are doing reasonably well to detect "message card". However, when the scene becomes complicated (e.g., the second row), our methods can detect more objects since more visual concepts can be extracted from our rich caption as queries for the detector. Similarly, it can be seen that our method captures more items that the AltText missed, e.g., "banana", "eggs", "butter", etc in the third row; "drawstring" in the fourth row; "apples" and "vases" in the last row. Also, it is more likely to extract a more detailed description of the region rather than a class name, such as "green-roofed cottage nestles" in Figure 3 and "decorative metal tree sculptures" in the image in the last row of Figure A. We believe such high-quality region labels essentially contribute to better supervision for CLOC pre-training.

## C MORE DISCUSSIONS

**Limitations.** One limitation for CLOC is the labeling efforts in preparing the training data. As we discussed in Section 1, there are no public large-scale region-text datasets since it is expensive to infer such labels up to the scales we consider here. Unlike previous work (Zhong et al., 2022) that cropping boxes from images for annotating, our VESL inference in *image-level* thus the cost does not scale with the number of detected regions. With that being said, such inference still requires hundreds of GPUs running in parallel for days to scale up to billions of images. We are working on releasing the annotations to accelerate future research for the community.

For CLOC, we focus on the training objective and framework formulation, while making minimal efforts on hyper-parameter tuning, architecture search, dataset cleaning, and etc., thus better performance could be achieved. Besides, although we have included extensive standard evaluation tasks, the fine-grained region knowledge could also be useful on more other under-explored tasks.

**Future directions.** We suggest promising future directions. In Section 3.2, our `Prompter` formulation can take flexible prompts to guide the embeddings for specific tasks. In this work, we consider a prompt as a single bounding box or a text caption, but it has the potential to expand to various types such as points, a mask, users' free-form referring, or multiple prompts in multiple types together. We think a more versatile `Prompter` with co-designs for different objectives can have a big potential. Similarly, our VESL labeling pipeline limits to detection box format. Annotators supported for more formats may further boost it. We believe our approach is promising, as more attention has been drawn recently for better re-captions (Li et al., 2024; Fan et al., 2024) that VESL relies on. In addition, CLOC model provides a new capability to extract region features without further training, and thus can be used as a foundation model for exploring new VL applications.

**Training cost.** We comment on the computation cost of our framework. Our large models (ViT L/14) were trained on 1024 v5p TPUs for about 6 days. To optimize Equation 2, CLOC needs extra computation. The main overheads come from the contrastive matrix but not the lightweight `Prompter`. Fortunately, we found it feasible since (1) only a few boxes in each image need to be sampled per update; (2) the loss computation becomes a smaller proportion when the ViT scales up. Overall, we found the computation acceptable compared to CLIP. More memory-efficient optimization like SigLIP (Zhai et al., 2023) can be implemented with JAX `shard_map`[8] ops.

**Discussions on design rationals.** Besides the main discussions we have stressed in the main text, here we provide more thoughts behind our design rationales that a reader may be wondering.

---

[8]https://jax.readthedocs.io/en/latest/jep/14273-shard-map.html

*(1) Why not use a local-enhanced encoder?* We would like to note that many encoders with great localization like DINOv2 (Oquab et al., 2023), OWLv2 (Minderer et al., 2024), CLIP-Self (Wu et al., 2023), etc. are developed specifically for dense vision tasks that cannot perform image zero-shot tasks like CLIP and CLOC. We would like to emphasize that our goal is to build a drop-in-replacement of CLIP encoder with better localization, without sacrificing CLIP's original capabilities such as image zero-shot tasks and its important backbone position for MLLMs. Furthermore, perhaps well-known within the MLLMs community, these encoders have been shown in recent reports that they are not comparable enough to compete with CLIP as the vision backbone for MLLM tasks (Tong et al., 2024) due to CLIP's superiority in vision-language alignment. We thus believe enhancing CLIP itself is more demanding as this paper focuses on.

*(2) Why not just train a CLIP with object detection?* One may wonder why we do not just train an encoder with joint optimization of the CLIP contrastive loss with some object detection loss instead of the CLOC design of Equation 4.

Although it sounds like a plausible approach, we would like to point out that contrastive pre-training and object detection are fundamentally quite different in their technical rationales. CLIP pre-training is often on large batches of low-resolution and noisy images, while object detection is trained on small batches of high-resolution images. CLIP is by default trained from scratch and object detection is typically initialized from pre-trained encoders and focuses on the detection head. Furthermore, detection requires heavy computation on box proposals to detect all boxes appearing in an image, while our region-text contrastive design allows us to flexibly sample fewer regions per image as motivated in Equation 3. Overall, their data pipeline and distributed training setup are not on the same scale thus such joint training may not be very reasonable.

With that being said, some previous works do have attempts that are the exceptions but only for some but not all of the mentioned aspects, and mainly for the purpose of detection. For instance, DetCLIP-v2 (Yao et al., 2023b) adds image-text contrastive loss into detection loss to improve open-vocabulary capability for detection. OWLv2 pre-trains the detector with rather small resolutions but still with a batch size of a maximum 256 since each image will need to predict up to 100 boxes during training. Both DetCLIP-v2 and OWLv2 fine-tune from a pre-trained encoder.

On the contrary, we study pre-training the encoder from scratch, which may be complementary to the previous efforts. CLOC maximizes the similarity in co-design with CLIP, thus making it much easier to develop within the same codebase.

*(3) Do we really need to train* CLOC *from scratch? What if we fine-tune from CLIP?* As CLIP pre-training is expensive, one may wonder if it is necessary to train from scratch on the proposed region-text datasets, or if we can initialize from a standard CLIP trained on image-text pairs only and fine-tunes with CLOC for a shorter stage. Our early investigation, even with extensive hyper-parameter tuning, suggests it is likely to be suboptimal compared to training from scratch directly. For instance, we initialize from the CLIP model ② in Table 2 and fine-tunes it for another extra 100K steps with the CLOC training loss Equation 4. The model reaches $64.1\%/19.1\%$ mAcc on COCO/LVIS region recognition, which is much worse than $70.1\%/27.2\%$ of the trained-from-scratch model ⑧, even with more overall training steps.

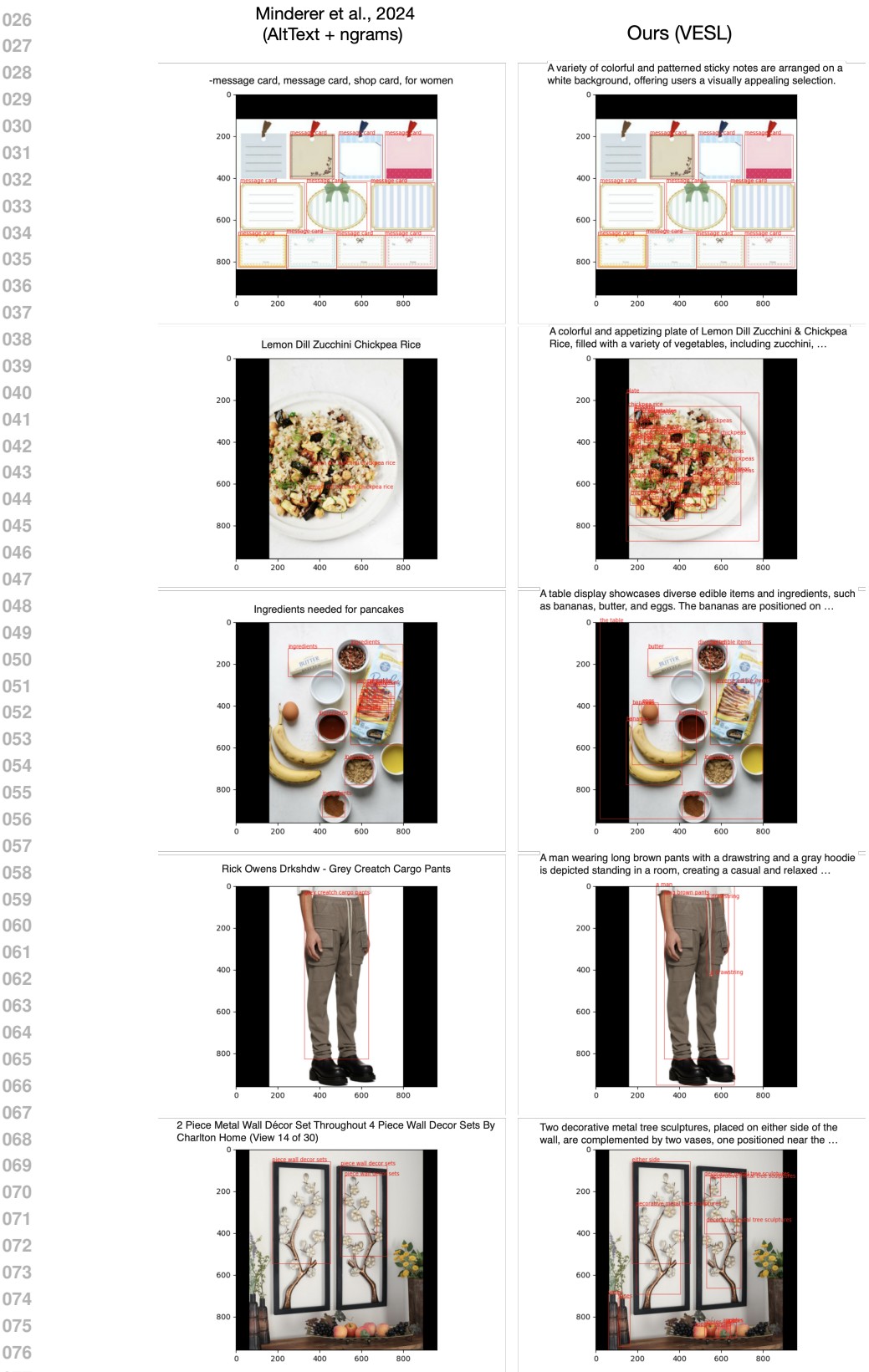

Figure A: Examples comparing our VESL and the labeling approach in (Minderer et al., 2024) that directly uses the $n$-grams of the crawled AltText. For VESL, each image is annotated with the visual-enriched caption to replace the AltText, which is used to generate region text candidates that capture the image content better.

