# OpenReview forum: "Contrastive Localized Language-Image Pre-Training"
_ICLR.cc/2025/Conference — Submitted to ICLR 2025_

### Official Review · Reviewer_MadS · 2024-10-23

**Soundness:** 3
**Presentation:** 3
**Contribution:** 1
**Rating:** 1
**Confidence:** 5

**Summary:**

The paper proposes to combine ITC with region text prediction. The authors show that a data labelling pipeline at scale for region proposal combined with a image-text dataset leads to model outperforming CLIP baseline for MMLLM among other tasks.

**Strengths:**

The author's model is well engineered to take latency into context while adding additional blocks.
The data engine uses synthetic captions rather than alt-text to generate data for the box predictor which is lesser explored.
The authors show improvement in multiple tasks over CLIP.

**Weaknesses:**

This paper would have been strong if other works did not explore better pretraining over CLIP for open vocabulary computer vision tasks. However, that is not true anymore. This work ignores all the advancement in the field of pretraining image-text representation models.
Several works in combining SSL with ITC exist and show better localization performance e.g. MaskCLIP and SILC. Moreover, BRAVE at ECCV2024 showed that these works also improve MMLLM performance.
Moreover, the primary idea of this work has also been explored in "LocCa: Visual Pretraining with Location-aware Captioners" recently.

While I commend the authors for doing an extensive study of their method vs CLIP, we have a different landscape today. It is not fair to claim large improvements without putting other improvements from the field in context. Moreover, while the authors improve upon CLIP, their performance metrics on imagenet or LVIS fall short compared to other works from the field. Therefore I unfortunately have to recommend a rejection for this paper.

**Questions:**

No questions, the work is very easy to read. I commend the authors for that.

---

> ### Author Response · Authors · 2024-11-24
> **Review response (1/2)**
>
> Thank you for taking the time to review our paper and provide your feedback.
>
> We appreciate your comments, but we believe there may be a significant misunderstanding. Please note that we have made every effort to ensure that every word and claim in our manuscript is both ethical and true to the principles of our research community. Please find our response below:
>
>
> * “This paper would have been strong if other works did not explore better pretraining over CLIP for open vocabulary computer vision tasks. However, that is not true anymore.”
>
> We would like to clarify that our work is fundamentally different from open-vocabulary computer vision tasks. Our primary contribution lies in enhancing the localization capability of the CLIP encoder, transforming it into a robust representation for downstream tasks, particularly MLLM tasks. Additionally, to support large-scale pre-training, we developed a visually enriched and spatially localized captioning framework to effectively generate region-text pseudo-labels at scale. We demonstrate that CLOC enables the creation of high-quality regional embeddings for image region recognition and retrieval tasks. Furthermore, it serves as a drop-in replacement for CLIP, significantly enhancing MLLMs, especially in referring and grounding tasks.
>
> We believe the distinction between our work and open vocabulary CV tasks should be evident. We also want to emphasize that novelty does not necessarily mean creating something entirely new. Introducing a new idea, examining existing approaches from a fresh perspective, or developing a method that advances the current state of knowledge can all qualify as novel contributions. We proposes a new encoder pre-training goal, promptable embedding, that is more suitable for the downstream MLLM. We design a prompter architecture on top of CLIP that enforces the image encoder to generate image embedding that is easy to be transformed into a region representation given a visual prompt, thus more suitable for MLLM to leverage. This perspective for improving CLIP pre-training for MLLM is novel.
>
>
> * “This work ignores all the advancement in the field of pretraining image-text representation models.“
>
> We respectfully disagree with the assertion that we are ignoring others' related work. We discuss the related work in the related work section (Section 2). To summarize, we appreciate the reviewer to list some work here, some of them are indeed related and we are happy to discuss more in the future version of the paper. However, we believe the reviewer has a serious misunderstanding of the concepts and our work. Our goal is very different than the listed work above and unique. As stated in the paper, we first improve MLLM by enhancing image encoder pre-training with better localization (Lines 49-52). Second, we explore how effectively pseudo-labeling images can scale up to region-level contrastive learning (Lines 53-86). We thank you for mentioning other works in the review. We will include a more comprehensive discussions on the additional related works in the next revision.
>
>
> * “Several works in combining SSL with ITC exist and show better localization performance e.g. MaskCLIP and SILC“
>
> We are unsure why the reviewer has linked our work to SSL (assuming it is self-supervised learning), as the two are entirely orthogonal. We acknowledge that combining SSL with ITC can potentially also improve localization performance; however, our work is not about combining SSL with ITC. As noted in Line 53: “we explore a data-driven approach that complements the original CLIP image-text pre-training objective with explicit region-text supervision.” Our method does not prevent adding another SSL objective with ITC, but in this work, we mainly focus on combining CLIP with region-text supervision with a scalable re-captioning framework.

---

> ### Author Response · Authors · 2024-11-24
> **Review response (2/2)**
>
> * “Moreover, BRAVE at ECCV2024 showed that these works also improve MMLLM performance.”
>
> We believe the reviewer may have misunderstood the concept of BRAVE.  The BRAVE paper does not make an argument that these encoders alone improve MLLM performance. In their Table 1, they only “benchmarked” existing public checkpoints, which “differ in terms of objective, training data, and model size” as noted in the caption of their Table 1 as well. BRAVE demonstrates that consolidating features from multiple existing frozen encoders into a more versatile representation can achieve superior performance. Their results show that MLLM performance can be significantly enhanced when better representations contain signals from diverse datasets and training objectives. This insight strongly aligns with our work, which focuses on incorporating region-level signals into image-level contrastive learning to further improve MLLM performance.
>
>
> * “Moreover, the primary idea of this work has also been explored in "LocCa: Visual Pretraining with Location-aware Captioners" recently."
>
> Thank you for pointing out the LocCA paper. While it is indeed related, we do not understand why the reviewer believes that the primary idea has already been explored in LocCA. The two models are fundamentally different. LocCA incorporates region-level (location-aware) information into a captioning objective, whereas CLOC primarily focuses on utilizing it within a contrastive objective. Additionally, the methods for constructing training data and the evaluation strategies differ significantly between the two approaches.
>
>
> * “While I commend the authors for doing an extensive study of their method vs CLIP, we have a different landscape today. It is not fair to claim large improvements without putting other improvements from the field in context.”
>
> We respectfully disagree with this statement. We did not claim large improvements over all other works from the field but only a fair comparison to CLIP, with the same images and training setup. Previous works have mainly explored either SSL objectives due to lack of large-scale datasets (Lines 75-80) or different models than CLIP. However, as for the MLLM community, CLIP is still arguably the default encoder choice (Line 36-49, Lines 978-981). We humbly believe our work actually fills in a missing piece and can be complementary to the current pre-training landscape.
>
>
> * “Moreover, while the authors improve upon CLIP, their performance metrics on imagenet or LVIS fall short compared to other works from the field.”
>
> Again, we respectfully disagree. We believe our performance is competitive in the same setup. For instance, regarding ImageNet accuracy with B/16, OpenAI’s CLIP reports 68.3%, SigLIP [ICCV 2023] reports 76.0%, MetaCLIP [ICLR 2024] reports 70.8%, and our model achieves 75.5%. As for LVIS, we did not find zero-shot region classification accuracy explicitly reported in other works, and our focus is not on LVIS detection. Please refer to the paragraph starting at Line 806 for additional details. But more importantly, the primary goal of our method is to enhance region-level metrics rather than image-level metrics, as we stated in the beginning.

---

> > ### Comment · Reviewer_MadS · 2024-11-26
> > **Lack of comparative analysis and baselines**
> >
> > I thank the authors for their response, however, it is very unfortunate that they are adamant on not comparing and attributing proper previous works. I urge the authors to treat the review process as a scientific screening process where we need to hold works to the highest standard and demand fair comparisons to existing works. This is needed to understand if the method provides "yet another training method" or if it actually contributes to the improvements in the field by offering new insights and/ or better performance.
> >
> > - "Our primary contribution lies in enhancing the localization capability of the CLIP encoder, transforming it into a robust representation for downstream tasks, particularly MLLM tasks."
> >
> > This goal has been studied by several works including the ones I had quoted in my response. However the authors still some how feel like their goal is unique and they are the first one to study it since the original CLIP paper.
> >
> > - "We also want to emphasize that novelty does not necessarily mean creating something entirely new. Introducing a new idea, examining existing approaches from a fresh perspective, or developing a method that advances the current state of knowledge can all qualify as novel contributions."
> >
> > I am personally a fan of simple ideas that are well experimented. However, we are a research community and we need to let the evidence speak for itself. The authors have ignored all previous works and have not done a fair comparison. In the research community, we care about how one algorithm compares to the other given several dimensions including data, compute and algorithmic complexity. The authors' work is not the first one in this topic and for this work to be useful to the research community, we need to know how it compares to methods beyond CLIP in all the key dimensions. I quoted BRAVE because it showed how different pretraining methods serve MMLLMs. Several of these methods are also aiming to have better localization features. Which according to authors' own words makes it a baseline to be compared with.
> >
> > * "We respectfully disagree with the assertion that we are ignoring others' related work. We discuss the related work in the related work section (Section 2)." "Our goal is very different than the listed work above and unique."
> >
> > I again respectfully disagree with the authors. Their goal is not unique and this is a well studied field. When you are proposing a new algorithm in a sub-topic, citing works is not enough. We need fair baseline comparisons.
> >
> > * "We are unsure why the reviewer has linked our work to SSL (assuming it is self-supervised learning), as the two are entirely orthogonal."
> >
> > Again, you are proposing a way to improve localization and MMLLM performance, this can be done several ways as shown in the literature. SSL is one approach to do it. Brave and other works has shown this.
> >
> > * "We believe the reviewer may have misunderstood the concept of BRAVE. The BRAVE paper does not make an argument that these encoders alone improve MLLM performance."
> >
> > I have not misunderstood your work as I have mentioned in my answer above. Brave brings several pretraining methods to the MMLLM field.
> >
> > * "Thank you for pointing out the LocCA paper. While it is indeed related, we do not understand why the reviewer believes that the primary idea has already been explored in LocCA. "
> >
> > Your base idea is that localization prior in pretraining helps in better representation learning. Your approach is an orthogonal way to implement this idea. A training baseline comparing the two would be beneficial to understand the benefit of your approach.
> >
> > - "However, as for the MLLM community, CLIP is still arguably the default encoder choice (Line 36-49, Lines 978-981). We humbly believe our work actually fills in a missing piece and can be complementary to the current pre-training landscape."
> >
> > This is again not true. While CLIP is arguably the most popular choice in the open source community, there have been several improvements since as I have mentioned several times.
> >
> > - "Again, we respectfully disagree. We believe our performance is competitive in the same setup."
> >
> > SigLIP from 2023 achieves 76% and SILC from 2024 achieves 76.6% zeroshot on imagenet. Your method is lacking by a whole point in zero-shot imagenet. Moreover, both Owl-v2 and SILC report zero-shot localization accuracy on LVIS at 29.2 and 32.4 respectively. CLOC's 28.5 is not competitive compared to them.

---

> > > ### Author Response · Authors · 2024-11-28
> > > **Response (1/2)**
> > >
> > > We thank the reviewer for their prompt response, which we find MUCH more helpful than the initial feedback suggesting a lack of novelty in our work and an oversight of prior research. However, we regret that the reviewer continues to dismiss our contributions and maintains their perspective despite the details we have provided.
> > >
> > > We want to stress that the scope of the paper is just to show our approach is a valuable improvement on CLIP, as we state in Line 105 “... we demonstrate that CLOC significantly and consistently outperforms the CLIP counterpart.” We did not overstate that our method is the best way among any methods except CLIP to improve localization. Instead, for those works based on SSL mentioned by the reviewer, our method is complementary to them as we are interested in a pseudo-labeling approach that was not studied before, as we motivated in Line 53. These encoders have been rained with different objectives and labels, which are expected to have different pros and cons. We think a more proper way to put different encoders into the same context is like BRAVE pointed out by the reviewer to ensemble different encoders together, rather than replacing one over others.
> > >
> > > With that being said, we understand the reviewer’s request for comparing our encoder with more other methods in a broader family of encoders such as the mentioned MaskCLIP, SILC, and LocCA. This is certainly a study nice to have but way over the scope of this paper, and should deserve another study paper in our humble opinions. These works are trained with different architectures, image datasets, computation, goals, etc. For instance, MaskCLIP is fine-tuning CLIP on COCO panoptic masks; SILC trains CLIP encoders with an extra teacher encoder for self-distillation and LocCA trains a pair of encoder and decoder, which are more expensive to train. Both SILC and LocCA are trained on the private WebLI dataset. Therefore, a comparison on their public checkpoints will not be fair since it will no be apple-to-apple comparisons due to different training data, architectures, hyperperameters, etc., and reproducing all of them will be a significant workload at this point thus we did not include such an experiment.
> > >
> > > Upon our explanation, we hope the reviewer can understand why we opt to focus on comparing with CLIP but not make a huge claim over the literature. We think a focused extensive comparison of our method with CLIP (as the reviewer originally agreed) can be a good contribution to the community without other confounders. In fact, this is the same practice in the SILC paper, where the authors compare their SILC with CLIP/SigCLIP counterparts on the same training setups but not against all the other methods.

---

> > > > ### Author Response · Authors · 2024-11-28
> > > > **Response (2/2)**
> > > >
> > > > To further address the reviewer’s other concerns, we try to summarize the points the reviewer has made:
> > > >
> > > > 1. We ignored all prior work, improving image representation from different encoders has been explored, e.g. BRAVE.
> > > > 2. Localization has been explored before for better representation learning, such as LocCa.
> > > > 3. We need to be competitive to SILC, SigLIP, and Owl-v2 on their corresponding metrics.
> > > >
> > > > We make further clarification below:
> > > >
> > > > **Response to Brave**
> > > >
> > > >
> > > > * “I quoted BRAVE because it showed how different pretraining methods serve MMLLMs. ... Brave brings several pretraining methods to the MMLLM field.”
> > > >
> > > > We thank the review for the clarification. This is helpful and we acknowledge the potential benefits of multiple encoders, as explored in works like Brave, COMM [1], Ferret-v2 [2], and many others. However, such approach can significantly increase latency and system complexity as they bring more components in. Our work aims to address this challenge by exploring ways to embed more information into a single encoder through unified contrastive learning, without significantly increasing model size. That said, we are still unclear why the reviewer believe this approach contradicts our contributions.
> > > >
> > > > [1] From CLIP to DINO: Visual Encoders Shout in Multi-modal Large Language Models
> > > > [2] Ferret-v2: An Improved Baseline for Referring and Grounding with Large Language Models
> > > >
> > > > **Response to LocCa**
> > > >
> > > > *  “Your base idea is that localization prior in pretraining helps in better representation learning. Your approach is an orthogonal way to implement this idea. A training baseline comparing the two would be beneficial to understand the benefit of your approach.”
> > > >
> > > > We thank the reviewer for acknowledging our approach is an orthogonal way to integrating the localization prior into a better representation. We fully agree that including such a comparison would help in understanding these two different approaches. However, we argue that this is not strictly necessary, as the approaches are orthogonal (also pointed by the comment here). In our work, we focused on contrastive learning, and our experiments have demonstrated the effectiveness of our approach. Additionally, it is worth noting that LocCa has not released their model or checkpoints, making a fair comparison even more challenging. Nonetheless, we are happy to make our best effort to implement this comparison and include it in the next version of the paper.
> > > >
> > > > **Response to SILC and Owl-v2**
> > > >
> > > >
> > > > * “SigLIP from 2023 achieves 76% and SILC from 2024 achieves 76.6% zeroshot on imagenet. Your method is lacking by a whole point in zero-shot imagenet.”
> > > >
> > > > We want to clarify SigLIP and SILC are training on the proprietary WebLI 10B image dataset that are not public available. Their ImageNet accuracy is 1 point higher does not imply our method is worse but primary because of different training data. We have compared CLIP on the same training data in our experiments for various tasks.
> > > >
> > > > * “Moreover, both Owl-v2 and SILC report zero-shot localization accuracy on LVIS at 29.2 and 32.4 respectively. CLOC's 28.5 is not competitive compared to them.”
> > > >
> > > > Again, as we said in the previous response, what Owl-v2 and SILC reports are the average precisions on the LVIS detection evaluation after fine-tuning their models on LVIS training set. Ours in Table 2 is zero-shot region classification task to evaluate the localization capability in the ablation study.  That is given a bonding box, the model needs to predict the class, similar to ImageNet image classification but for regions. They are completely different tasks and metrics. Please refer to the paragraph starting at Line 806 for additional details.
> > > >
> > > > **Misc:**
> > > >
> > > > *  “While CLIP is arguably the most popular choice in the open source community, there have been several improvements since as I have mentioned several times.”
> > > >
> > > >
> > > > Based on the summary above, we believe the "several improvements" the reviewer mentioned are orthogonal and actually high motivates our work. We believe a paper improving on a focused dimension with careful ablation should to be considered as a contribution to community.

---

> > > > > ### Comment · Reviewer_MadS · 2024-12-02
> > > > >
> > > > > I thank the authors for taking my comments more openly this time. However, I don't have more information from the authors to change my rating.
> > > > >
> > > > > The authors mention that all baselines are trained with different datasets. This is again why I feel strongly that we need to show how a pretraining method stacks up with baseline literature. SILC for example showed their method's performance on a comparable dataset to open-source and with the previous SOTA on their internal dataset in their supplementary. This instills confidence that a certain baseline is pushing the research community forward.
> > > > >
> > > > > I am not against your work. As I mentioned before, I like simple well executed ideas. However, we need consistent baseline comparisons to know how a certain work pushes the research forward. I believe your work has potential but it is not ready for the research community yet. Your extensive training setup shows that you have the budget to train some baselines for fair comparisons. I believe that can benefit the community and help us judge the work's merits.
> > > > >
> > > > > Regarding OWL-v2 performance, the numbers I quoted are zero-shot. If you check their paper again, their numbers are higher when finetuned on LVIS.

---

> ### Author Response · Authors · 2024-12-04
>
> We thank the follow-up response from the reviewer. After the fruitful discussion, we respect the reviewer’s arguments. However, we still respectfully disagree with the unreasonably negative rating on our paper. We believe a rigorous ablation against CLIP to prove the effectiveness of a simple approach is valuable to be presented to the research community. This is not against the larger scope mentioned by the reviewer to consider either comparing to or combining with another encoder pre-training method from a broader family, which we will keep investigating in the future.
>
> We have some points to further clarify. We do have the budget to train more models but for a fair comparison, we need to re-implement or reproduce the baseline methods since our implementation is in JAX (as mentioned in section 5.1). Also, for many works including MaskCLIP and SILC, we have not found their training implementation to be released. For the numbers of LVIS on our paper and OWL-v2 paper, again as we stressed, they are both zero-shot but for completely different tasks and metrics. Ours in Table 2 is zero-shot region classification, while OWL-v2 paper reports open-vocabulary detection average precision. The numbers cannot be compared.

---

### Official Review · Reviewer_FruC · 2024-11-02

**Soundness:** 3
**Presentation:** 3
**Contribution:** 3
**Rating:** 6
**Confidence:** 3

**Summary:**

This paper enhances the region-level representation in CLIP and introduces a new concept termed "promptable embeddings." The region-level representation is a critical task in this domain. The experimental results appear promising.

**Strengths:**

1. This paper addresses a significant challenge: enhancing the region-level representation of CLIP, which is crucial for various applications, including multi-modal large language models (MLLMs).
2. The experimental results presented in this paper demonstrate promising performance across both image-level and region-level tasks.

**Weaknesses:**

Several related works are not adequately addressed in the manuscript, such as 'UMG-CLIP: A Unified Multi-Granularity Vision Generalist for Open-World Understanding' (ECCV 2024).

**Questions:**

Will the collected fine-grained region-text annotations be made publicly available? This aspect is highlighted as one of the main contributions of the study.

---

> ### Author Response · Authors · 2024-11-22
> **Thanks for your review!**
>
> We thank you for your positive review and questions!
>
> For UMG-CLIP, we did have mentioned it in footnote 1 (Line 269). At the time of the submission (9/28), it is shown as a withdrawn arXiv preprint and was later updated on 10/29. We are more than happy to add UMG-CLIP as an official reference and include more discussions about related works in the final version. UMG-CLIP uses a region-level contrastive loss similar to our Equation 2. Also, their model extracts region embedding by a RoI Align layer, which we actually have compared as the baseline for our prompter design (discussion in the paragraph “Extracting region embedding with visual prompts” in Line 252). We have ablation experiments to support it — the second item of observations of Table 2 around Line 421 and in row 3&4 in Table 3. Besides, UMG-CLIP is fine-tuned from CLIP on its 41M image datasets, which our work considers scaling up to 2B images (see Table 1) and pre-training from scratch.
>
> About the region-text annotations, as mentioned in the reproducibility statement, we are working hard on releasing the annotations with internal legal approvals for the public.

---

### Official Review · Reviewer_NXpc · 2024-11-03

**Soundness:** 2
**Presentation:** 3
**Contribution:** 2
**Rating:** 6
**Confidence:** 3

**Summary:**

Contrastive image-text pretraining has achieved significant success in recent years; however, low-quality annotations and the limitations of contrastive learning alone remain challenges. This work aims to enhance CLIP by incorporating fine-grained annotations, enabling the model to scale to billions of data points and demonstrating substantial improvements over the CLIP baseline.

CLOC can be viewed as a practical approach to enhancing dataset quality, with a particular focus on fine-grained grounding details.

**Strengths:**

1. The data refinement process is scalable to billions of samples, demonstrating its practical utility in vision-language pretraining.

2. The approach is intuitive and straightforward, leading to improved grounding capabilities.

3. The architecture is easily adaptable for downstream grounding tasks, enhancing versatility.

4. This method effectively handles large-scale datasets (billions-level in this work), with comprehensive experiments conducted despite significant computational demands.

5. In addition to widely used recaption, this work also include region phrase candidates extraction and open-vocabulary detection with extracted phrases.

**Weaknesses:**

1. Visually-enriched and spatially-localized models, such as QWEN, rely on robust training to achieve high-quality outputs. These models contribute essential knowledge to the process, with the quality of grounding pseudo-labels also largely dependent on these off-the-shelf models.

2. Significant work has already been done to improve dataset captions, as seen in studies like *Improving Multimodal Datasets with Image Captioning* by Thao et al. (NeurIPS 2023). Integrating object grounding information into vision-language pretraining is also an established concept, exemplified by X-VLM (Zeng et al., ICML 2021). Although CLOC’s approach to grounding differs somewhat from X-VLM, these differences are minor and do not detract from the overall contribution.

**Questions:**

1. What explains the significant improvement that CLOC achieves over LLaVAW and MME-C? In contrast, GQA, which heavily relies on precise object grounding, shows only limited improvement.

---

> ### Author Response · Authors · 2024-11-22
> **Thank you for the review!**
>
> [W1]
>
> We agree that the labeler is always important for pseudo-labeled data. The goal of our work is indeed to explore how to effectively scale up the training data (the paragraph from line 82). The focus of the proposed visually-enriched and spatially-localized pipeline in section 4 is how to best leverage off-the-shelf models for region-text pre-training data. We think our report is a positive motivation for both the grounding, captioning (also will be discussed in [W2]), and CLIP pre-training community, as we show that a proper combination of them can benefit our CLOC pre-training.
>
> [W2]
>
> We want to clarify that we do not claim the contributions of using better image captions to improve multimodal tasks like Thao et al. and the VeCap paper we followed (line 322, line 451). In fact, our report embraces the advance of better captioning for MLLMs. As we noted in the future directions (line 958), we think our work provides another motivation for researchers to develop better re-captioning, which we think should be considered a strength rather than a weakness of our paper.
>
> For X-VLM, we thank you for providing the reference and we will update our paper. X-VLM is an early pioneer in studying combining several losses like contrastive learning, matching, and masked language modeling by training on only 16M images. In contrast, we scale up pseudo-labeled data up to 2B images. In terms of integrating regional information into training, we want to point out that CLOC’s design does make a significant difference. X-VLM extracts regions from the image feature by the corresponding spatial patches similar to RoI pooling, which we have compared as a baseline approach of our prompter design discussed in the paragraph “Extracting region embedding with visual prompts” (line 252). As discussed in the second item of observations of Table 2 around Line 421, our prompter design is more effective in the scale we considered, observing 12-27% region task accuracy difference compared to the RoI baseline. In the Ferret experiments (row 3&4 in Table 3), pre-training with RoI-Align is worse than CLOC prompter by 4.6% on Ferret VQA tasks.
>
> [Q1]
>
> Good question. CLOC improves on CLIP with region embedding training thus region classification/retrieval (Table 2) and referring and grounding VQA tasks (Table 3, 4) are more direct evaluations. For general VQA, the performance gains are somewhat indirect and depend on many factors. We hypothesize LLaVAW and MME are relatively new benchmarks that are more challenging in targeting out-of-domain generalization and more diverse in terms of visual concepts, while GQA is constructed based on Visual Genome that was widely studied and may be better considered already in the SFT data of the LLaVA models.

---

> > ### Comment · Reviewer_NXpc · 2024-12-03
> >
> > Thank you for your detailed feedback.
> >
> > My questions regarding GLOC have been thoroughly addressed.
> > While the dataset collection heavily relies on off-the-shelf VeCap, I recognize the value of the recaptioning approach, particularly given its application at a billion-scale level. Additionally, the rebuttal has partly alleviated my concerns about similarities with prior works like X-VLM. The evidence showing that pre-training with RoI-Align performs worse than CLOC is compelling.
> >
> > However, despite differences in implementation, the core ideas of re-captioning and grounding in MLLM domain are already established in related works, which significantly diminishes the novelty and impact of this study.
> >
> > This paper is well-executed and highly polished, but not interesting. I will maintain my current score.

---

### Official Review · Reviewer_LJyF · 2024-11-04

**Soundness:** 3
**Presentation:** 3
**Contribution:** 3
**Rating:** 5
**Confidence:** 4

**Summary:**

This paper addresses a critical limitation of CLIP by introducing CLOC (Contrastive Localized Language-Image Pre-training), which enhances CLIP's capability to understand region-level semantics while maintaining its strong image-level performance. The key contributions include: (1) a novel concept of "promptable embeddings" that allows easy transformation of image embeddings into region representations, (2) a lightweight Prompter module that effectively extracts region features without compromising CLIP's original capabilities, and (3) a scalable pseudo-labeling pipeline (VESL) that generates high-quality region-text annotations. The approach is validated through extensive experiments across 31 evaluation tasks, showing particular effectiveness in MLLM applications.

**Strengths:**

1. Strong practical value:
    - Addresses a real limitation in CLIP that affects its use in MLLMs
    - Provides a drop-in replacement solution without compromising original capabilities
    - Demonstrates consistent improvements across various downstream tasks
2. Technical novelty:
    - Introduces an elegant concept of "promptable embeddings"
    - Develops a lightweight and effective Prompter module (though it reminds me of prompt encoder of SAM[^1])
    - Proposes a scalable approach to generate region-text annotations
3. Comprehensive evaluation:
    - Extensive experiments across 31 tasks
    - Thorough ablation studies
    - Strong performance improvements in MLLM applications

[^1]: Kirillov, Alexander, et al. "Segment anything." Proceedings of the IEEE/CVF International Conference on Computer Vision. 2023.

**Weaknesses:**

1. Insufficient comparison with related works:
    - Missing comparison with DenseCLIP[^1] and similar region-aware CLIP variants [^2, ^3, ^4, ^5]
    - While RegionCLIP[^6] is mentioned, the differentiation from other similar approaches is not clearly articulated
    - Limited discussion of trade-offs compared to existing solutions
2. Missing implementation details:
    - The region sampling strategy (how 4 regions per image are selected) is not specified
    - The rationale for setting λ as the ratio of images with region labels in the mini-batch is not explained
    - The impact of batch composition (ratio of images with/without region labels) on training is not discussed
    - The threshold (0.9) for similar text filtering lacks justification
    - The definition and extraction method of "leaf entities" in NER is unclear
    - Computational overhead analysis of the Prompter compared to RoI pooling is missing
3. Limited analysis of scalability:
    - The saturation of performance at 300M images for region tasks needs more investigation
    - The impact of the threshold for similar text filtering on training time is not analyzed (in addition to (10) in table 2)

[^1]: Rao, Yongming, et al. "Denseclip: Language-guided dense prediction with context-aware prompting." Proceedings of the IEEE/CVF conference on computer vision and pattern recognition. 2022.
[^2]: Gu, Xiuye, et al. "Open-vocabulary object detection via vision and language knowledge distillation." arXiv preprint arXiv:2104.13921 (2021).
[^3]: Zhou, Xingyi, et al. "Detecting twenty-thousand classes using image-level supervision." European Conference on Computer Vision. Cham: Springer Nature Switzerland, 2022.
[^4]: Feng, Chengjian, et al. "Promptdet: Towards open-vocabulary detection using uncurated images." European Conference on Computer Vision. Cham: Springer Nature Switzerland, 2022.
[^5]: Lin, Jiayi, and Shaogang Gong. "Gridclip: One-stage object detection by grid-level clip representation learning." arXiv preprint arXiv:2303.09252 (2023).
[^6]: Zhong, Yiwu, et al. "Regionclip: Region-based language-image pretraining." Proceedings of the IEEE/CVF conference on computer vision and pattern recognition. 2022.

**Questions:**

1. Region Sampling Strategy
    - How are the 4 regions per image selected during training?
    - Is there any consideration for overlap between regions?
    - How does the sampling strategy affect the model's performance?
2. Training Implementation
    - What is the rationale behind setting λ as the ratio of images with region labels?
    - How do you handle the batch composition in terms of images with/without region labels?
    - Have you experimented with different similar text filtering thresholds besides 0.9?
3. Scalability and Efficiency
    - Why does the performance saturate at 300M images for region tasks but continue to improve for MLLM tasks?
    - How does the computational cost of Prompter compare to traditional RoI pooling approaches?
    - What is the memory overhead of storing region embeddings compared to traditional CLIP?

---

> ### Author Response · Authors · 2024-11-22
> **Thank you for the review!**
>
> Thank you for the detailed feedback and questions!
>
>
> * Related works.
>     * We thank you for providing many related works and we will incorporate them into the next version. We want to emphasize that all these works [^1 ^2 ^3 ^4 ^5 ^6] are improving for open-vocabulary (OV) object detection. Our goal is not to build a detection or grounding model but to pre-train a better CLIP encoder for MLLM tasks.  As said in our discussions of RegionCLIP you have also pointed out (the paragraph from line 183 and the “Improving localization of CLIP” paragraph in section 2), that OV detection focuses on recognizing many objects and labeling them from a large vocabulary set. VQA focuses on holistic reasoning across the whole image via an MLLM. Also, as we discussed in the appendix “Why not use a local-enhanced encoder?” (line 972), these works mainly leverage CLIP for the OV detection task only. Our goal is to increase CLIP pre-training with region-level understanding without sacrificing the CLIP original image-language capability, which is crucial for MLLM tasks.
>     * We believe these works are quite different and complementary from ours in terms of their methodology. We provide a concise summary here. DenseCLIP [^1] fine-tunes CLIP on a supervised pixel loss (e.g., with a segmentation dataset) but not changing CLIP pre-training. [^2 ^3 ^4 ^5] are mainly leverage CLIP as an external (frozen) classifier for expanding standard detection into open-vocabulary detection.
> * We leave the answers for the implementation details and scalability to the same questions below
> * About the the leaf entities in NER, we mean the noun phrases detected by the NER parser as shown in the example in the lower boxes in Figure 3. We provide the codes in Listing 1 in the Appendix. We will make this more clear in the final version.
> * Questions about the region sampling strategy
>     * The 4 regions are sampled randomly during training.
>     * We adopt the OWLv2 detector that was trained with DETR-style loss [1], which naturally suppresses overlapping regions with the same object. Therefore, we do not specifically add assumptions on such cases.
>     * We ablate sampling with 2 regions per image in item 12 in Table 2 and found sampling more (M=4) is slightly better.
> * Questions about training Implementation
>     * The intuition of \lambda we set is to make the additional CLOC loss (weighted by \lambda in Equation 4) proportional to the ratio of images that have region labels. For instance, if there are 80% of the images in a mini-batch have region labels, \lambda should be 0.8.
>     * The images without region labels will only contribute to the CLIP loss but not the CLOC loss and grounding loss.
>     * We did lightly tune the threshold in [0.5, 0.8, 0.9 0.95].
> * Questions about scalability and efficiency
>     * The region tasks in Table 1 are constructed mainly for ablation and development purposes (line 412-413). We think the zero-shot region tasks performance saturates since the tasks are relatively simple in terms (e.g., classifying from some fixed set of classes from COCO and LVIS).  MLLM tasks require a higher level of understanding which might benefit from a larger scale of data.
>     * Compared to RoI pooling, the Prompter is a lightweight single-layer self-attention layer. The extra computation cost is negligible compared to the whole forward-backward.
>     * As discussed in the paragraph of Line 962, the extra overhead of the region embeddings is mainly for the contrastive loss as M times more examples, while this is an acceptable constant especially when training with a larger ViT. Note that, the main overhead is the VIT encoding and each image is only encoded once but not M times, which is the reason we design the prompter on top of the ViT (Line 245-247).
>
> [1] End-to-End Object Detection with Transformers, 2020

---

> > ### Comment · Reviewer_LJyF · 2024-11-26
> >
> > Thank you for your detailed responses to my comments and questions. I appreciate the clarifications you provided regarding the region sampling strategy, training implementation, and scalability aspects of your work. Your explanations have addressed many of my initial concerns about the implementation details and the reasoning behind certain design choices. It's also encouraging to hear that you plan to incorporate the related works I mentioned into your revised version to provide a more comprehensive context for your contributions.
> >
> > However, after further reflection and considering Reviewer MadS's comments, I have growing concerns about the novelty and positioning of your work relative to existing literature. It appears that several recent studies, such as MaskCLIP, SILC, and BRAVE, have explored similar ideas in enhancing CLIP's localization capabilities and improving performance on MLLM tasks. The absence of direct comparisons to these methods makes it challenging to assess the true impact and advancement your approach offers over the current state of the art. Additionally, given that your performance metrics do not significantly surpass these existing methods, it becomes difficult to justify acceptance without a thorough evaluation against them. In light of these considerations, I am adjusting my rating accordingly.

---

> > > ### Author Response · Authors · 2024-12-04
> > >
> > > We thank the reviewer for acknowledging that we have addressed the initial questions. We are more than happy to incorporate the related works mentioned by the reviewer into the final version.
> > >
> > > Regarding the comparison to MaskCLIP, SILC, and BRAVE, we want to emphasize that our contributions are orthogonal and complementary to their works. More specifically, our work studies a different approach in comparison with these works:
> > >
> > > * CLOC vs. MaskCLIP [Dong et al., CVPR 2023]: MaskCLIP is a self-training approach that combines CLIP contrastive loss with masked distillation on the image/text tokens. Our approach is scaling up region pseudo labels to 5B images. They train on YFCC-15M and report 44% ImageNet zero-shot accuracy, which is much worse than ours (76% accuracy). In addition, MaskCLIP has much more complicated architectures which keeps an additional teacher model and a decoder for both image and text on top of CLIP. MaskCLIP requires much more memory and training cost and was trained with a 4096 batch size, while our model is more efficient which only adds a lightweight prompter and we trained with a 32K batch size.
> > > * CLOC vs. SILC: Similar to MaskCLIP, SILC is another self-training method combining CLIP with DINO self-distillation loss. They do not study how pseudo-labeling can help CLIP as a weakly-supervised approach. In addition, their models are trained on the private WebLI dataset thus direct comparison to their released checkpoint is not very meaningful.
> > > * CLOC vs. BRAVE: BRAVE demonstrates the benefits of using features from multiple independent encoders, as also explored in works like COMM [1], Ferret-v2 [2], and many others. However, such an approach can significantly increase latency and system complexity as they bring more components in. Our work aims to encode more information into a single encoder through unified contrastive learning, without significantly increasing the overall number of parameters.
> > >
> > > Overall, these methods and our CLOC model all complement CLIP from different perspectives and can be combined together to further improve. For a conference paper, our paper pivots a focused research problem with extensive studies under fair and comparable conditions rather than universally pursuing SOTA numbers. We hope our clarification addresses the reviewer’s new concern and we sincerely hope the reviewer could consider restoring the positive score.
> > >
> > > [1] From CLIP to DINO: Visual Encoders Shout in Multi-modal Large Language Models
> > >
> > > [2] Ferret-v2: An Improved Baseline for Referring and Grounding with Large Language Models

---

### Meta-Review · Area_Chair_q2zq · 2024-12-20

**Metareview:**

This paper proposes a pretraining method to enhance the localization ability of contrastive image-text pretraining (i.e., CLIP), which is considered an important topic in the research field as CLIP models are widely used across various applications. The paper presents a new methodology with experimental results across different model sizes (ViT-B/L/H) and tasks. The paper's strengths are its strong motivation, clear writing, and comprehensive experiments.

However, as Reviewer MadS and Reviewer LJyF pointed out, the paper lacks conceptual and experimental comparison with relevant previous works (e.g., MaskCLIP, SILC, BRAVE, etc.). While the reviewers and the AC understand that these previous works may differ in method direction and experimental settings, the AC believes that it would be valuable to compare them as baselines to demonstrate the advantages of the proposed method within the broader context of CLIP localization ability.

Given the paper's notable strengths, the AC recommends submitting this work by addressing these comparison aspects to future conferences/journals rather than this ICLR.

**Additional Comments On Reviewer Discussion:**

The AC agrees with Reviewer MadS that it is necessary to compare this work with existing research focused on improving localization ability in image-text pretraining.

---

### Decision · Program_Chairs · 2025-01-22

Reject